



# Atmospheric aerosols in Rome, Italy: Sources, dynamics and spatial variations during two seasons

**Caroline Struckmeier[1], Frank Drewnick[1], Friederike Fachinger[1], Gian Paolo Gobbi[2] and Stephan Borrmann[1, 3]**

[1]{Particle Chemistry Department, Max Planck Institute for Chemistry, Mainz, Germany}

[2]{Institute of Atmospheric Sciences and Climate, ISAC-CNR, Roma, Italy}

[3]{Institute for Atmospheric Physics, Johannes Gutenberg University, Mainz, Germany}

Correspondence to: C. Struckmeier (c.struckmeier@mpic.de), F. Drewnick (frank.drewnick@mpic.de)

## Abstract

Investigations on atmospheric aerosols and their sources were performed during October/November 2013 and May/June 2014 subsequently in a suburban area of Rome (Tor Vergata) and in central Rome (near St. Peter's Basilica). During both years a Saharan dust advection event temporarily increased $PM_{10}$ concentrations at ground level by approximately $10\ \mu g\ m^{-3}$. Generally, during Oct/Nov the ambient aerosol was more strongly influenced by primary emissions, whereas higher relative contributions of secondary particles (sulphate, aged organic aerosol) were found during May/June. Absolute concentrations of anthropogenic emission tracers (e.g. $NO_x$, $CO_2$, particulate polyaromatic hydrocarbons, traffic-related organic aerosol) were generally higher at the urban location. Positive matrix factorisation was applied to the $PM_1$ organic aerosol (OA) fraction of aerosol mass spectrometer (HR-ToF-AMS) data in order to identify different sources of primary OA (POA): traffic, cooking, biomass burning, and (local) cigarette smoking. While biomass burning OA was only found at the suburban site, where it accounted for the major fraction of POA (18-24 % of total OA), traffic and cooking were more dominant sources at the urban site. A particle type associated with cigarette smoke emissions, which is associated with a potential characteristic marker peak ($m/z$ 84, $C_5H_{10}N^+$, a nicotine fragment) in the mass spectrum, was only found in central Rome, where it was emitted in close vicinity to the measurement location. Regarding secondary OA, in Oct/Nov, only a very aged, regionally advected oxygenated OA was found,



which contributed 42-53 % to the total OA. In May/June total oxygenated OA accounted for 56–76 % of the OA. Here also a fraction (18-26 % of total OA) of a fresher, less oxygenated OA of more local origin was observed. New particle formation events were identified from measured particle number concentrations and size distributions during May/June 2014 at both sites. While they were observed every day at the urban location, at the suburban location they were only found under favourable meteorological conditions, but independent of advection of the Rome emission plume. Particles from sources in the metropolitan area of Rome and particles advected from outside Rome contributed 42-70 % and 30-58 % to total measured $PM_1$, respectively. Apart from the general aerosol characteristics, in this study the properties (e.g. emission strength) and dynamics (e.g. temporal behaviour) of each identified aerosol type is investigated in detail in order to provide a better understanding of the observed seasonal and spatial differences.

## 1 Introduction

Atmospheric aerosol particles remain a major uncertainty in both, estimations of climate change (Boucher et al., 2013) and of impact of air pollution on public health (Heal et al., 2012), and therefore are a major topic of current research (Fuzzi et al., 2015). Identifying the sources, properties and concentrations of atmospheric particles is essential for evaluating their effect on climate and health and constitutes a crucial step in finding measures for the improvement of air quality.

Many studies on aerosols and their sources have been performed in urban environments (e.g. Freutel et al., 2013; Mohr et al., 2012; Zheng et al., 2005), which are characterized by high population densities and a large diversity of particle sources. Typical urban aerosol sources include road traffic, cooking, and heating activities. Also emissions from biomass burning can be important, both of regional origin (e.g., agricultural and wild fires; Reche et al., 2012), and from residential wood combustion, which recently has become more prominent in Europe even in urban environments (Fuller et al., 2013).

Many of these anthropogenic sources emit large amounts of organic material in the fine particle fraction (e.g. Hildemann et al., 1991). In recent studies of particle source identification (e.g. Allan et al., 2010; Mohr et al., 2012; Reche et al., 2012), positive matrix factorisation (PMF) was applied to separate the organic aerosol (OA) fraction into different factors associated with various OA sources, thereby providing indications about the fraction





of primary and secondary organic aerosol (POA and SOA) (Zhang et al., 2011). Oxygenated
organic aerosol (OOA), mainly associated with SOA, is typically found to be the most
abundant fraction of OA (Lanz et al., 2010), with concentrations depending on season and
location (Zhang et al., 2011). Several studies, mainly such from observations during summer
time (Lanz et al., 2010), show discrimination of OOA into a fresher and a more aged type of
OOA based on different states of oxygenation and/or volatility (Jimenez et al., 2009).
While AMS measurements yield useful information on the age of OA, they cannot provide
evidence for new particle formation of fresh secondary aerosol. Indications for such, however,
can be found in physical aerosol properties like particle number concentration or size
distributions (e.g. Alam et al., 2003). New particle formation events in urban environments
have been investigated previously in several studies (e.g. Alam et al., 2003; Brines et al.,
2015; Minguillon et al., 2015; Shi et al., 2001; Zhang et al., 2004), and especially in the early
afternoon seem to be responsible for elevated particle number concentrations in urban areas in
Southern Europe (Reche et al., 2011).
On the other hand, while the health impact of coarse particles ($PM_{10}$-$PM_{2.5}$) is not yet fully
understood (Heal et al., 2012), the association between Saharan dust advections and
mortality/hospitalisation is quite well demonstrated (Stafoggia et al., 2016). Deserts are large
sources for mineral dust, which can strongly contribute to atmospheric $PM_{10}$ levels, especially
in Southern Europe. Measurements performed in the period 2001-2004 during Saharan dust
advections over Rome showed a mean Saharan dust contribution of 12-16 µg m$^{-3}$ to daily
$PM_{10}$ concentrations, leading to an average annual increase of about 2 µg m$^{-3}$ (Gobbi et al.,
2013). In the central Mediterranean region, maximum dust concentrations are typically
observed from spring to autumn (Barnaba and Gobbi, 2004).
In this study, we investigate the occurrence and properties of ambient aerosol from different
types of sources in Rome, which apart from local emissions can be influenced by advected
aerosol from continental Europe and the Sahara desert. During two different seasons (Oct/Nov
2013 and May/June 2014) and at two different locations (city centre and suburb), stationary
measurements of chemical and physical properties of aerosols, several trace gases, and
meteorological variables were performed. Non-refractory components of submicron particles
were measured with an Aerodyne high-resolution time-of-flight aerosol mass spectrometer
(HR-ToF-AMS). To support identification of particle sources, their strength and temporal
behaviour, the OA measured with the HR-ToF-AMS was further separated into different
factors using PMF.



Based on these measurements, in this work the urban atmosphere of Rome is investigated in
terms of particle source identification with a special focus on seasonal and spatial differences
influencing the presence and/or the characteristics of aerosol types in the city area.
## 2 Experimental
### 2.1 Measurement locations and periods
Measurement results presented in this study were obtained during four intensive field
campaigns in the greater Rome area, Italy (Table 1). The city of Rome covers an area of
$1300 \text{ km}^2$ and has a population of about 2.9 million residents (about 4.3 million residents
within the whole metropolitan area of $5350 \text{ km}^2$). Three airports are located in the Rome
province, including the largest one in Italy (Fiumicino). Heavy industries are not found in
Rome; the economy is mainly based on services, education, construction, tourism, etc. Parks
and gardens cover some 34 % of the city area. Rome is characterised by high traffic volume
and density: about 50 % of the population commutes every day, mainly by private cars. The
cars per capita ratio in the city is 550 per 1000 inhabitants.
Measurements referred to as DIAPASON were performed during Oct/Nov 2013 and
May/June 2014 at the Institute of Atmospheric Sciences and Climate (CNR-ISAC) in Tor
Vergata, Rome. The institute is located in the south-eastern outskirts of Rome
(41°50'30.2''N, 12°38'51.2''E, 103 m a.s.l., 14 km from central Rome) and considered an
urban background site. The measurement platform MoLa (see Sect. 2.2) was positioned at a
free field with no buildings within a radius of 200 m. A frequently used street is located at
approximately 100 m distance in northern direction. The closest highway (A1) is situated
south-westerly at a distance of about 700 m. Single-house villages are scattered over this
territory, starting some 1 km from the site. Frascati, a town on the Alban Hills, is located at
about 4 km distance in south-easterly direction. During both periods measurements at Tor
Vergata were supported by the EC-LIFE+ project DIAPASON (Desert-dust Impact on Air
quality through model-Predictions and Advanced Sensors ObservatioNs), which aims on
improving existing tools to assess the contribution of Saharan dust to local $PM_{10}$ levels
(http://www.diapason-life.eu/, last access 09.05.2016). For this reason measurements were
scheduled in periods where a dust advection event could be expected and was forecasted by a
number of dust forecasts such as the DREAM8b (Basart et al., 2012), the SKIRON (Kallos et
al., 1997) and the Tel Aviv University (Alpert et al., 2002) models.



The "POPE" (Particle Observations around St. PEter's) measurement campaigns were
conducted during November 2013 and June 2014 in central Rome. Measurements were
performed inside a courtyard belonging to the administration of the hospital "Santo Spirito"
(41°54'04.3"N, 12°27'41.5"E, 18 m a.s.l.), which is positioned approximately 600 meters from
St. Peter's Basilica. This urban measurement site is surrounded by highly frequented streets,
separated from the courtyard by the four-storey building of the hospital. The surrounding area
is a touristic hotspot with frequent religious gatherings (e.g. festivals, masses) and many
restaurants and shops. Especially on Wednesdays during the papal audience and on Sundays,
if the masses are held at St. Peter's or during papal speeches (Angelus), the area attracts
numerous visitors.
The distance between the two measurement sites is around 17 km. During both years
measurements at Tor Vergata and central Rome were performed subsequently.

## 2.2 Instrumentation

All measurements were performed with the **Mo**bile aerosol research **La**boratory MoLa
(Drewnick et al., 2012). MoLa is based on a regular Ford Transit delivery vehicle equipped
with instruments for on-line measurements of chemical and physical properties of aerosols,
important trace gases and meteorological variables (Table 2). Further description as well as
details of the aerosol inlet system can be found in Drewnick et al. (2012). All results
presented in this study were obtained in stationary measurements, with the aerosol inlet and a
meteorological station at 7 m above ground level.
An HR-ToF-AMS (Aerodyne Research, Inc.; DeCarlo et al., 2006) was used to measure
particulate mass concentrations of submicron non-refractory organics ("Org"), sulphate
("$SO_4$"), nitrate ("$NO_3$"), ammonium ("$NH_4$") and chloride ("Chl"). The HR-ToF-AMS
allows the distinction between different ions at the same nominal mass-to-charge-ratio ($m/z$).
The instrument was run in V-mode, i.e. the ions followed a "V"-shaped trajectory through the
mass spectrometer, allowing high sensitivity at slightly lower mass resolution, compared to
the higher resolution mode (W-mode).
In the framework of the EC-LIFE+ project DIAPASON additional measurements were
performed at the Tor Vergata measurement site, which aimed at assessing the contribution of
Saharan dust to PM levels. These measurements included hourly $PM_{10}$, a three-wavelength





nephelometer, one-hour filter sampling for off-line PIXE analysis (Lucarelli et al., 2014) and
a polarization LIDAR-ceilometer for the assessment of presence, phase and altitude of aerosol
layers (Gobbi et al., 2004). Boundary layer heights were determined from polarisation
LIDAR-ceilometer measurements based on the method described by Angelini and Gobbi

5    (2014).

Since the POPE measurements were performed inside a courtyard, surrounded by four-storey
tall buildings, wind speed, wind direction and solar radiation data are affected and not used
for these periods. The time resolution for all measurements is 60 s or better.
**3   Data preparation and analysis**
**3.1   General data analysis**
All measured variables were corrected for sampling delays and set on a common 1-second
time base. Particle losses during the transport through the inlet system were negligible
(Drewnick et al., 2012). The data time series were carefully inspected and quality checked.
Data affected by instrument calibrations and malfunctions were removed. Measurement
periods influenced by local emissions (e.g. moving vehicles in the immediate vicinity of
MoLa) were identified based on prominent short peaks in the time series of $CO_2$ and particle
number concentration (PNC) which significantly exceeded the typical variability, and
removed from the data set. After data decontamination, 5-minute averages were calculated for
all variables, which were used for all following analyses if not otherwise indicated.
Data collected during the DIAPASON2013/POPE2013 and DIAPASON2014/POPE2014
field campaigns are presented in local winter (UTC+1) and local summer (UTC+2) time,
respectively. For convenience, DIAPASON2013 data are presented only in winter time, even
though the change from summer to winter time was at the fifth day of measurements
(27.10.2013). This means data measured prior to the time change is 1 hour shifted to the past
with respect to local (summer) time. Especially diurnal patterns dominated by anthropogenic
activity patterns (e.g. traffic during rush hour times) could be affected by ignoring the time
change. In order to evaluate this possible influence, diurnal cycles measured before and after
the time change were compared, but no significant shift of diurnal patterns was observed
between the time periods. Since diurnal cycles are not only modulated by the source emission
strengths, but also by boundary layer dynamics, we assume the missing evidence of the time





shift in the data is caused by a combination of influence from boundary layer dynamics and
the temporal uncertainty of diurnal cycles calculated over only a few days. Additionally,
anthropogenic activities could have partially not been instantly adapted to the time change,
which would lead to a blurring of the effect of the time change on diurnal cycles.
Polar plots of species concentration as a function of local wind direction and wind speed were
generated by averaging species concentrations (60 s data) into bins of 5° wind direction and
0.5 m s$^{-1}$ wind speed. The resulting data were smoothed by applying a natural neighbour
interpolation (Sibson, 1981). As presented by Yu et al. (2004), such polar plots can provide
directional information on sources in the vicinity of a monitoring site. Sources close to the
measurement site are typically indicated by concentration decreases with increasing wind
speed, while pollutants which are emitted from remote sources or at higher altitudes need
higher wind speeds to be transported to the monitoring site (Yu et al., 2004). Similarly,
Carslaw et al. (2006) reported the capability of such bivariate polar plots to distinguish
between no-buoyancy sources like traffic (decreased pollutant concentration with increasing
wind speed) and buoyant plumes emitted from sources like chimney stacks (increased
concentrations with increasing wind speed), where the plume needs to be brought down to
ground-level from a higher altitude.

## 19   3.2   HR-ToF-AMS data analysis

AMS data evaluation was performed within Igor Pro 6.37 (Wavemetrics) with the standard
AMS data analysis software SQUIRREL 1.55H and PIKA 1.14H. Elemental ratios calculated
from organic ion fragments (Aiken et al., 2007) were determined using APES light 1.06 (all
available at http://cires1.colorado.edu/jimenez-group/ToFAMSResources/ToFSoftware/). For
all data sets a collection efficiency of 0.5 was applied, which is typical for the given ambient
measurement conditions (Canagaratna et al., 2007). The ionisation efficiency (IE) of the ion
source and the relative ionisation efficiency (RIE) for ammonium and sulphate (e.g.
Canagaratna et al., 2007) were determined before the DIAPASON and after the POPE
campaigns in both years. An additional IE calibration in 2013 after the field measurements
showed no general trend in IE values. Therefore, the observed variability of the IE values is
assumed to stem only from the uncertainty of the calibrations, and for each year averages of
the determined IE and RIE values were used for data analysis. Measurements of particle free





air were carried out multiple times during the campaigns and were used for correction of
instrumental background effects.
In order to separate total OA into different aerosol types, PMF (Paatero and Tapper, 1994;
Ulbrich et al., 2009) was applied to high-resolution mass spectra of the OA fraction with *m/z*
below 131. The procedure of HR data and error matrices preparation is described in detail in
DeCarlo et al. (2010). Isotopes constrained to a fractional signal of their parent ion were
excluded from the analysis. Within the PMF Evaluation Tool v2.06 ions with signal-to-noise
ratio < 0.2 were removed from data and error matrices, and ions with signal-to-noise ratio
between 0.2 and 2 were down-weighted in the analysis by increasing their estimated error by
a factor of two (Ulbrich et al., 2009). Particulate $CO_2^+$ (*m/z* 44) and its associated ions at *m/z*
16, 17, 18 and 28 were down-weighted by a factor of $\sqrt{5}$ (Ulbrich et al., 2009, supplemental
information).
In order to find the most reasonable and robust PMF solution, the number of factors (one up
to ten, always at least two more than the finally selected solution), the rotational force
parameter (fPeak: -1 to 1; $\Delta$ = 0.1) and the starting point (seed: 0 to 50; $\Delta$ = 1) were varied
(see Ulbrich et al., 2009 for methodological details). Solutions with fPeak=0 and seed=0
turned out to yield robust results for all data sets. The evaluation of potential PMF solutions
was based on comparisons of the resulting factor time series with those of co-located
measurements (see Sect. 4.2), and of factor mass spectra with such from the literature.
Residues, i.e. the contribution of organic mass concentrations not included in any of the
factors, accounted for <1 % of the organics mass concentration in all used PMF solutions and
are therefore negligible.

## 4   Results and discussion

### 4.1   Overview: Differences between seasons and locations

This section provides a broad overview of the mean conditions of local meteorology and air
quality during each measurement campaign (Table 3, Fig. 1), and discusses their seasonal and
spatial differences. Figure 2 provides an overview of the relative composition of non-
refractory $PM_1$ plus BC and shows the contribution of different factors related to different
sources retrieved from the OA fraction using PMF. In total, seven different factors were
identified: OOA (oxygenated OA), SV-OOA (semi-volatile OOA), LV-OOA (low-volatile





OOA), HOA (hydrocarbon-like OA), COA (cooking OA), BBOA (biomass burning OA) and
CSOA (cigarette smoke OA, see Sect. 4.2.5; considered local contamination and not included
in the pie charts in Fig. 2). Here, only a general overview of these different factors focusing on
seasonal and spatial differences is given. A more detailed discussion of the various sources
associated with these factors is provided in Sect. 4.2.
*Meteorology overview:*
The first period of the DIAPASON2013 campaign (23.-31.10.2013) was dominated by high
pressure influences and low wind speeds with air masses moving from the Atlantic across the
Mediterranean basin. Within this period dust from the Sahara was transported to the Rome
area (see Sect. 4.2.1). The second half of DIAPASON2013 (1.-7.11.2013) was characterized
by a pressure drop and increased wind speed together with some frontal passages leading to
precipitation. Both turbulent kinetic energy (TKE), which is a measure of the intensity of
turbulence and can be used as an indicator for the mixing efficiency of pollutants in the air
(Srivastava and Sarthi, 2002), and boundary layer heights (BLH) were at rather low levels
during DIAPASON2013 (Table 3), favouring the accumulation of pollutants.
The first measurement days of POPE2013 (07.-09.11.2013) were influenced by changing
weather conditions, followed by a low pressure system centred over Italy (10.-14.11.2013)
driving N-NW wind conditions and leading to almost daily precipitation events. High TKE
levels (almost three times higher compared to DIAPASON2013) and slightly increased
boundary layer heights (900 m compared to 850 m; Table 3) led to conditions were dilution of
pollutants was more favoured.
During DIAPASON2014 a low pressure system was located over the Atlantic and North
Africa. Saharan dust was advected to the measurement site during the first week of
measurements. Some precipitation events occurred during these advections. TKE was slightly
increased after the dust advection, favouring the reduction of pollutant concentrations.
Boundary layer heights reached around 1500 m (Table 3).
During POPE2014 a period of low pressure over the Atlantic and high pressure over Africa
and Europe (04.-13.6.2014) was followed by reversed conditions (14.-17.6.2014) with some
heavy precipitation events. Compared to DIAPASON2014 the boundary layer was slightly
higher (1560 m) and TKE was decreased by 20 %.





During DIAPASON2013 local winds were predominantly arriving from south and south-
easterly directions, whereas during DIAPASON2014 south-westerly wind directions
dominated (Fig. 1; no data available for POPE2013/2014, see Sect. 2.2). No clear relationships
of air mass origin with measured $PM_1$ mass concentrations were found by the calculation of
HYSPLIT (Stein et al., 2015) and FLEXPART (Stohl et al., 2005) backward trajectories for
our measurement periods.
*Aerosols and trace gases:*
During both, Oct/Nov and May/June measurements, $CO_2$, $NO_x$ and particulate PAH
concentrations (all typically traffic-related) were higher in central Rome compared to the
outskirt location (Table 3). Cooking-related aerosol (COA) concentrations were found to be
generally higher in the city centre (Fig. 2b, d), while traffic-related (HOA) concentrations
were strongly increased during 2014 (+ 58 %) at the urban location (Fig. 2d), and nearly the
same at both locations during the 2013 measurements (Fig. 2a, b). All this is consistent with
increased primary emissions (cooking, traffic) at the urban (Fig. 2b, d) compared to the
suburban site (Fig. 2a, c). However, total concentrations of organic aerosol predominantly of
primary origin (POA) were higher at the suburban (1.9-2.5 µg m$^{-3}$) compared to the urban
(1.4-1.6 µg m$^{-3}$) location. This is due to a factor indicating particles from biomass burning
(BBOA), which was obtained exclusively at Tor Vergata (Fig. 2a, c). Here, biomass burning
seems to be an important particle source, contributing the most abundant fraction (42-51 %)
of POA. Consistently, BC, which is related to primary emissions from both biomass burning
and traffic, showed no general trend between the two locations (Table 3).
The influence of increased temperatures and stronger solar radiation during measurements in
May/June 2014 (Table 3) is reflected in elevated ozone mixing ratios and the fact that it was
possible to extract an additional OOA factor, which was attributed to a fresher, less oxidised
aerosol type (SV-OOA, Fig. 2c, d). Within a continuum of OOA with different degrees of
aging/oxidation, SV-OOA (fresh, of rather local origin) and LV-OOA (aged, of more regional
origin) are located in the upper and lower range, respectively. In contrast, during Oct/Nov,
only one type of rather aged OOA was found (Fig. 2a, b), due to reduced photochemistry in
this season which prevents the quick formation of oxygenated aerosol from precursors emitted
in the vicinity.
A stronger influence of aged aerosol of rather regional origin on the $PM_1$ fraction was
observed for May/June than for Oct/Nov: the fraction of OOA (SV-OOA + LV-OOA) to OA,





but also the relative contribution of total organics and sulphate to $PM_1$ were higher in the
warmer season (Fig. 2c, d). This could be caused by different prevalent air mass origins,
though the analysis of backward trajectories provided no definite answers on this (see above).
Also the relative fraction of sulphate could be lowered in Oct/Nov due to an enhanced fraction
of primary particles as a consequence of lower boundary layer heights (Table 3) and,
potentially, higher emission strength of local sources during the colder season. Consistently,
in Oct/Nov 2013 at both measurement locations a higher BC fraction was observed compared
to May/June, and POA made up a larger fraction of the total measured organics (Fig. 2). Also
absolute BC concentrations were enhanced. Additionally, higher concentrations of $NO_x$ and
PAH (increased by a factor of 3), and of total particle number concentration (PNC) were
observed during Oct/Nov (Table 3), indicating the accumulation of pollutants during the
colder season due to the aforementioned reasons.
*"Home-made" vs. "advected" $PM_1$:*
For a rough estimate of the contribution of $PM_1$ originating from sources in the Rome
metropolitan area and from advection from outside, $PM_1$ species were separated into "home-
made" (BC, HOA, COA, BBOA, SV-OOA, $NO_3$) and "advected" (OOA/LV-OOA, $SO_4$)
(Table 4). $NH_4$ was apportioned to home-made and advected $PM_1$ based on the molar
concentrations associated with $NO_3$ and $SO_4$, respectively. Not included in these estimates are
dust advection periods and emissions from cigarettes (which were considered as local
contaminations). During the measurements the fraction of home-made $PM_1$ accounted for 42-
70 % and advected $PM_1$ accounted for 30-58 % of total (home-made + advected) $PM_1$
(Table 4). During the 2013 measurements higher absolute concentrations of home-made $PM_1$
were found at the suburban location compared to the urban site, possibly caused by
meteorological conditions favouring pollutant accumulation during the respective period (see
above). During 2014, when meteorological conditions (e.g. BLH, TKE; see Table 3) were
rather comparable at both locations, similar absolute concentrations of home-made $PM_1$ were
observed at both sites. No general seasonal difference in home-made $PM_1$ fraction was
observed, although BLH was strongly increased during the May/June 2014 compared to the
Oct/Nov 2013 measurement periods. Partially, this might be due to the additional presence of
the home-made species SV-OOA during May/June, which could to some extent have
compensated for the dilution effect due to the increased BLH. Altogether, neither a general
spatial (DIAPASON vs. POPE) nor a seasonal (Oct/Nov vs. May/June) tendency regarding
the contribution of home-made and advected $PM_1$ to total $PM_1$ was observed. These results





indicate that urban air quality in Rome strongly depends on both, emissions within the city
and transport of pollutants to the city, which both contribute to urban aerosol concentration.
Independent of measurement season and location the organics fraction always was found to
contribute the largest share of $PM_1$ (44-53 % of non-refractory $PM_1$ plus BC, Fig. 2), though
its relative composition (primary vs. secondary OA) differed. Regarding absolute $PM_1$
concentrations (Table 3 from EDM measurements, Fig. 2 from non-refractory components
plus BC) neither any general conclusion whether aerosol mass concentrations are higher at the
city centre or in the suburb, nor whether $PM_1$ concentrations are elevated during any of the
two different seasons, can be drawn. In the 2013 campaign total $PM_1$ mass concentrations
were more than doubled at the suburban compared to the urban location, whereas in the 2014
measurement $PM_1$ concentrations were increased by a factor of 1.4 at central Rome. As
discussed above, changes in meteorological conditions are likely one explanation for this
result: During DIAPASON2013 meteorological conditions favoured the accumulation of
pollutants, whereas the dilution of pollutants was favoured during the POPE2013
measurement period; during DIAPASON2014 and POPE2014 TKE and BLH were rather
similar leading to comparable pollutant dilution effects during the two measurement periods.
BLH were increased by around 75 % during May/June compared to Oct/Nov 2013, leading to
stronger dilution capacities in general. In addition to meteorological conditions (e.g. solar
radiation, BLH, TKE, air mass origin, etc.) local air quality can be strongly influenced by
local emission from various sources (traffic, cooking, biomass burning). A strong influence of
meteorological conditions (air mass origin) on air quality was also observed during the
MEGAPOLI campaign in Paris in July 2009, where variations in secondary aerosol
concentration mainly were attributed to such reasons (Freutel et al., 2013).

## 4.2   Aerosol sources: identification and characterisation

In this section the various aerosol types and sources which were identified from the data
obtained during the DIAPASON and POPE measurement campaigns in 2013 and 2014 are
discussed in more detail. Each identified aerosol type was characterized in an attempt to
determine its contribution to total particulate mass and its seasonal and spatial variability.
Furthermore, the potential origin of the identified aerosol types is discussed.





### 4.2.1 Saharan dust

During each DIAPASON field campaign one dust advection event lasting for several days was observed. The identification of the dust events with dust reaching down to the ground was based on dust forecasts provided by the SKIRON model (Kallos et al., 1997) and on co-located polarisation LIDAR-ceilometer measurements (Gian Paolo Gobbi, personal communication). PIXE analysis of 1-hour filter samples confirmed a significant increase of mineral dust concentrations at ground level during the identified dust periods (Silvia Nava, personal communication 2016). Table 5 provides the time intervals of "dust" and "no dust" (i.e. background with respect to dust advections) periods for DIAPASON2013 and DIAPASON2014. Based on these, estimations regarding the contribution of dust to PM levels were made by calculating the coarse particle fraction ($PM_{10-2.5}$) from EDM measurements for the respective periods (Table 5).

During DIAPASON2013 a Saharan dust advection period was observed from October 29[th] (5 pm) until November 1[st] (8 am) with $PM_{10-2.5}$ concentrations at ground level being increased by 78 % with respect to background conditions. Total $PM_{10}$ concentrations were elevated by 68 %, with 68 % and 70 % of mass belonging to $PM_1$ during the background and during the dust event periods, respectively. This increase in absolute $PM_1$ with a slight increase in the fraction of $PM_1$ during the dust advection compared to background conditions was also reflected in the particle size distributions (Fig. 3, black traces). The dust event was forecasted by the BSC-DREAM8b model (Basart et al., 2012; Perez et al., 2006a; Perez et al., 2006b) and also HYSPLIT (Stein et al., 2015) back trajectories showed dust transport from the Sahara with main dust sources located at 30-33N, between Morocco (Saharan Atlas) and SW Tunisia (Erg Oriental), in the period 26.-28.10.2013, moving along an anti-cyclonic pattern.

During DIAPASON2014 a dust event was observed from May 20[th] (1 pm) until May 26[th] (9 am). Here, the coarse particle fraction ($PM_{10-2.5}$) was increased by 180 % compared to the "no dust" interval with much smaller fractions of particle mass in $PM_1$ (31 and 48 % in the "dust" and "no dust" periods, respectively). The contribution of dust to the coarse particle fraction with almost no contribution to the submicron fraction is reflected in the difference particle mass size distribution (Fig. 3, bottom panel, red trace), which shows maximum mass concentrations for aerodynamic particle diameters around 2 µm and smaller contributions extending down to ~600 nm and up to more than 10 µm particle diameter. Comparing both dust events in terms of particle sizes, the contribution of the dust advection event during DIAPASON2014 was characterized by a broad particle mass size distribution with maximum





concentrations at particle sizes around 2 µm, whereas in 2013 two modes (with maxima
around 0.6 µm and 3 µm) were observed. Both the BSC-DREAM8b model and HYSPLIT
back trajectories showed dust transport from the Sahara to occur between 19.-22.05.2014
along a cyclonic pattern, with dust originating (as in October 2013) at 30-35N between
Morocco (Saharan Atlas) and SW Tunisia (Erg Oriental).
With regard to the coarse particle mode ($PM_{10-2.5}$) the dust event during DIAPASON2014 was
more distinct. In terms of absolute $PM_{10}$ concentrations, higher concentrations at ground level
were reached during the dust advection measured during DIAPASON2013. However, with
respect to the "no dust" conditions, mean contributions of approximately 10 µg m$^{-3}$ dust to
$PM_{10}$ concentrations were observed during the campaigns in both years, which is lower than
the mean increase of $PM_{10}$ in the order of 12-16 µg m$^{-3}$ observed by Gobbi et al. (2013)
during dust advection events in the period 2001-2004. During both advection events legal
$PM_{10}$ limits of the European Union (daily mean value of 50 µg m$^{-3}$) were not exceeded.
AMS data were investigated for potential impacts of Saharan dust advections on the chemical
composition of non-refractory submicron particles. Figure 4 presents the mean chemical
composition of non-refractory $PM_1$ measured during "dust" and "no dust" periods for both
DIAPASON campaigns. Only in 2013 minor differences in absolute mass concentrations
were observed, consistent with the slight increase of submicron particles observed in the mass
size distributions during this dust event (Fig. 3, bottom panel, black trace). However, the
relative composition remains the same for both periods in both years. This result suggests that
there was no significant influence of the dust advection on the chemical composition of the
submicron non-refractory aerosol fraction.
In order to cross-check whether differences in meteorological conditions could have biased
these results, and e.g. could have compensated for changes due to the dust affecting non-
refractory $PM_1$, averages of meteorological variables were calculated for "dust" and "no dust"
periods. The only slight differences found between dust events and background conditions
were in local wind directions (SE compared to S/SE in 2013, and SW compared to SSW in
2014) and (in 2013) in wind speeds (($1.4 \pm 0.7$) m s$^{-1}$ compared to ($2.0 \pm 1$) m s$^{-1}$). Since these
differences are only very minor, we conclude that our observation of comparable chemical
composition of non-refractory $PM_1$ during "dust" and "no dust" periods was not caused by
any compensating effects.



### 4.2.2 Secondary and aged aerosol: seasonal influence on formation processes and chemical composition

Seasonal variations of the characteristics of secondary and aged aerosols were identified by investigating new particle formation events and particle chemical composition measured during both POPE and DIAPASON campaigns in Oct/Nov 2013 and May/June 2014.

*New particle formation:*

Diurnal cycles of size-resolved and total particle number concentrations (PNC) for the Oct/Nov 2013 and May/June 2014 campaigns (Fig. 5) revealed several seasonal differences. During measurements in Oct/Nov 2013 (Fig. 5, left panels), elevated PNC were only observed during rush hour times in the morning and the evening, whereas in the measurements in May/June 2014, an additional PNC peak occurred with a maximum around 1-3 pm (DIAPASON2014) and 2-3 pm (POPE2014), respectively (Fig. 5, right panels). At the urban site (Fig. 5, bottom panels), PNC after the morning rush hour remain at an elevated level, likely because of increased background concentrations due to generally higher traffic density in the city centre. PNC were generally higher in the Oct/Nov 2013 than in the May/June 2014 campaigns, as discussed in Sect. 4.1.

Mean particle number size distributions for the period of maximum PNC at midday (Fig. 6; "nucleation"; solid traces) show a distinct mode at small particle diameters between 7-15 nm for both May/June 2014 campaigns (also visible in Fig. 5, right panels) compared to the number size distribution measured between 10-11 am (Fig. 6; "background"; dashed traces). Such occurrences of ultrafine particles at midday, when concentrations of particles from traffic are at a relative minimum and thus not responsible for strongly increased PNC, have been attributed before to new particle formation characteristic for urban areas with high solar radiation (Brines et al., 2015; Minguillon et al., 2015; Reche et al., 2011). During POPE2014 an additional mode at larger particle sizes ($D_p$ approximately 15-50 nm) was found in the particle size distribution measured at midday (Fig. 6), probably originating from increased background levels.

During POPE2014 diurnal cycles of mean (grey) and median (black) PNC agree very well with each other, also during the midday peak (Fig. 5). This reflects the observed low day-to-day variability during this period for the measurements in central Rome, wherein the corresponding PNC time series a midday peak was observed on every single day. Local smoking activities (see Sect. 4.2.5) at the central Rome measurement location seem not to





have biased these results, since no differences in diurnal cycles of PNC for weekdays
(smoking activities) and weekends (no smoking activities) have been found. This suggests
that the formation of new particles around midday was taking place every day at central Rome
during the May/June 2014 measurement period.
Conversely, the diurnal cycle of total PNC measured during DIAPASON2014 shows a clear
discrepancy between mean and median values during the midday peak (Fig. 5). This
discrepancy is due to the fact that new particle formation events did not occur on all days,
probably induced by different meteorological conditions and/or differences in pre-existing
particle surface areas (e.g. Kulmala and Kerminen, 2008).
To test whether particular meteorological conditions can promote/suppress new particle
formation events, "nucleation" and "non-nucleation" days were classified for
DIAPASON2014 by comparing PNC measured during 10-11 am (background conditions,
$PNC_{bg}$) and during 11 am - 4 pm (typical nucleation periods, $PNC_{nuc}$) (Table 6). This
classification was cross-checked by verifying if during classified nucleation days a clear
increase in PNC at small particle diameters ($D_p \leq 25$ nm) could be observed in the particle
number size distributions, and whether it was missing on classified non-nucleation days. Only
one potentially falsely classified nucleation day (24.05.2014) was found by checking these
criteria, and was moved to the class of non-defined days. The classification resulted in six
nucleation days, six non-defined days and two non-nucleation days.
Mean values for the time period 10 am to 4 pm (new particle formation period plus one
previous hour) were calculated for each day and averaged according to the above-mentioned
classification for the DIAPASON2014 campaign. Table 6 lists PNC of the classified periods
and variables potentially supporting new particle formation. A slight trend of increased
temperature, solar radiation and ozone levels and of low relative humidity characterizes
nucleation days compared to non-nucleation days and non-defined days. This is consistent
with previously reported association of high solar radiation (Pikridas et al., 2015; Shi et al.,
2001), low relative humidity (Kulmala and Kerminen, 2008) and increased ozone
concentrations (Harrison et al., 2000) with new particle formation events. No relationship
between the occurrence of new particle formation and the presence of Saharan dust was
observed. Different to the findings of Zhang et al. (2004), no increase of sulphate, ammonium
and nitrate concentrations was observed in our measurements during periods with new
particle formation events. Estimations based on the size distribution measurements during
DIAPASON2014 reveal that less than 1 % of $PM_1$ can be assigned to particles generated by





new particle formation. Therefore, it is not surprising that no significant influence of the
particle formation events on the AMS-measured chemical particle composition could be
observed.
In summary, our data do not provide sufficient statistical evidence to unequivocally determine
the driving factors for new particle formation. However, since indications for such were only
observed in the warmer season at both locations, it is probably linked to higher temperatures
and stronger solar radiation. Similar results were obtained from the MEGAPOLI
measurements in Paris, where new particle formation was only observed during summer
(Pikridas et al., 2015). During the May/June 2014 campaigns, new particle formation events
occurred roughly on 43 % of the measurement days at the suburban location, but on each
single day at central Rome, potentially due to increased concentrations of precursors and
higher prevailing mean temperatures (Table 3). In a long-term measurement study performed
by Costabile et al. (2010) the occurrence of aged nucleation mode particles (up to 30 nm) was
observed predominantly in spring in the area of Rome at a regional background site (located
more remotely than the Tor Vergata site) in the early afternoon (3 pm) when the measurement
site was located downwind of Rome (Brines et al., 2015). In contrast, during
DIAPASON2014 measurements, no dependency between nucleation events and wind
direction was observed, and the site was not located downwind of Rome during nucleation
periods. At this measurement location, probably less precursors are available than in central
Rome, but more precursors than at a remote regional background site such as in the study of
Costabile et al. (2010). This probably facilitates new particle formation events in the direct
vicinity of the site under favourable meteorological conditions, but independent of advection
of air masses from central Rome.
*Secondary and aged aerosol:*
Because of extremely low mass contributions from freshly formed particles to total particle
mass, new particle formation had no influence on the measured total organics mass
concentrations. However, a general seasonal difference in the composition of the oxygenated
organic aerosol (OOA) as determined in the PMF analysis was found, as discussed in the
following.
OOA, an aerosol type with increased oxygenation level, typically dominates the OA fraction.
It is assumed to be mainly formed in the atmosphere from gaseous biogenic and
anthropogenic precursors by photochemical oxidation, thus indicating SOA. Additionally,





some OOA may originate from atmospheric aging of POA. Generally, aging processes are
reflected in an increased degree of aerosol oxidation (Jimenez et al., 2009) leading to a larger
fraction of $m/z$ 44 ($CO_2^+$) in the aerosol mass spectra, generated by thermal decomposition of
carboxylic acids in the AMS (Ng et al., 2010). Additionally, a prominent peak in OOA mass
spectra occurs at $m/z$ 43 ($C_3H_7^+$ and $C_2H_3O^+$). Under conditions where sufficient freshly
oxidised organic aerosol is available in the ambient air, PMF can separate the OOA into two
factors associated with mass spectra containing different relative fractions of $m/z$ 44 and $m/z$
43 (f44 and f43, ratio of $m/z$ 44 and $m/z$ 43 signal, respectively, to the total signal of organics):
(I) A less oxidised, fresher, more locally produced semi-volatile OA (SV-OOA) associated
with higher f43, and (II) a stronger oxidised, more aged low-volatile OA (LV-OOA)
associated with higher f44 (Ng et al., 2010).
During our measurements the relative contribution of OOA to total organics varied between
42-76 %, with a slight increase during the warm period (Fig. 2). However, the main seasonal
difference was found in the composition of the OOA fraction. During the Oct/Nov 2013
campaign only one type of OOA was found, whereas for May/June 2014 PMF analysis
resulted in two OOA-factors: SV-OOA and LV-OOA. This observation is typical for
conditions with higher temperatures and stronger solar radiation, associated with increased
photochemistry (Jimenez et al., 2009), which enhance secondary aerosol formation and aging
processes and therefore facilitate the separation of OOA into fractions of different aging
levels. Similar observations were made during the MEGAPOLI measurements in Paris, where
only one factor describing OOA was identified in winter (Crippa et al., 2013a), whereas
during summer SV-OOA and LV-OOA could be separated (Crippa et al., 2013b).
The ratio of f44 to f43 gives an indication on the mean aging level of the aerosol (Fig. 7; Ng
et al., 2010). LV-OOA (from DIAPASON and POPE 2014, red markers) and OOA (from
DIAPASON and POPE 2013, green markers) fall into the same region in the f44 vs. f43 plot
(Fig. 7), indicating similar aging stages. SV-OOA (from DIAPASON and POPE 2014, blue
markers) shows a much higher fraction of $m/z$ 43 together with a decreased $m/z$ 44 fraction,
which suggests a low-oxidised, less aged particle type. Also the recombined "LV-OOA+SV-
OOA" (black markers in Fig. 7) shows a stronger contribution of f43 compared to OOA,
indicating an overall higher fraction of less oxidised organic aerosol in the warmer season.
We assume that LV-OOA (and OOA) is mainly advected and consists of strongly processed
material, whereas the low oxidation level of SV-OOA suggests a fresh, more locally produced



aerosol which was quickly formed from regional precursors as a consequence of increased
photochemistry during this season.
This hypothesis is tested by the use of polar plots, which connect species concentration with
information on local wind direction and speed, thereby indicating the origin of a certain type
of aerosol (see Sect. 3.1). Figure 8a shows the colour coded concentration of SV-OOA, LV-
OOA, $NO_3$ and $SO_4$ depending on wind direction and speed obtained during the
DIAPASON2014 measurements. SV-OOA concentrations are increased during low wind
speed conditions, indicating nearby, no-buoyance sources. In contrast, LV-OOA
concentrations are almost independent of wind speed with only slightly increased
concentrations during periods of high wind speed with mainly south-westerly wind directions
(direction of Tyrrhenian Sea). This suggests that LV-OOA is not associated with sources
located in the vicinity of the measurement site, but long-range transported to the site e.g. over
the ocean or from central Europe. The polar plot characteristics of $NO_3$, which is often used as
a tracer for semi-volatile aerosol (DeCarlo et al., 2010; Lanz et al., 2007), show strong
similarities to the ones of SV-OOA (Fig. 8a). Also $SO_4$ and LV-OOA show polar plot patterns
similar to each other (Fig. 8a), confirming the characteristics of an aged, regionally
transported aerosol.
The polar plot of OOA obtained for DIAPASON2013 shows increased concentrations
particularly during periods of north-easterly, but also during south-westerly wind directions
(Fig. 8b). During conditions of low wind speed, OOA concentrations are increased
independent of the prevailing wind direction. In contrast to the findings for DIAPASON2014,
for this data set similar polar plot characteristics as for OOA were observed partly in the plots
of $NO_3$, $NH_4$ and $SO_4$ (Fig. 8b). Elevated $NH_4$ and $SO_4$ concentrations were mainly measured
during times with south-westerly wind direction, whereas $NO_3$ was rather advected from
north-easterly direction. Based on the polar plot characteristics no consistent trend indicating
the aging level, the source or the formation process of the OOA fraction can be observed,
consistent with the assumption of advection of a rather aged type of OOA together with
different amounts of $NO_3$, $SO_4$ and $NH_4$ depending on air mass history.

### 4.2.3 Particles from biomass burning

The type of primary organic aerosol at Tor Vergata identified from PMF analysis which had
the largest share during both measurement periods was attributed to biomass burning





(biomass burning OA, BBOA). BBOA was identified by comparison with the time series of
known ion fragments of levoglucosan ($C_3H_5O_2^+$ ($m/z$ 73) and $C_2H_4O_2^+$ ($m/z$ 60); Schneider et
al., 2006) and by correlating the BBOA mass spectra with those presented by Mohr et al.
(2012). With Pearson's $R^2$ = 0.57-0.59 rather poor correlations were obtained, which
demonstrates the complexity and the potential variations of the BBOA mass spectra due to
aging processes and differences in source processes (e.g. different burning conditions or fuels;
Weimer et al., 2008). The mean BBOA mass concentration was 1.28 µg m$^{-3}$ (24 % of total
OA) during Oct/Nov 2013 and 0.82 µg m$^{-3}$ (18 % of total OA) during May/June 2014,
respectively (Fig. 2). Increased BBOA concentrations during the colder season probably result
from stronger agricultural burning activities (green waste burning) and domestic heating, as
well as from lower boundary layer heights. In the evening of 25.10.2013 during a strong,
visually detectable biomass burning event, which could be related to green waste burning in
the nearby Alban Hills, maximum concentrations of BBOA were obtained (up to 75 µg m$^{-3}$).
This event was used during the identification of the PMF solution: Only a factor including this
event could be considered to be attributed to biomass burning emissions.
The origin of BBOA emissions was further investigated by relating BBOA mass
concentrations to local wind direction and speed (see Sect. 3.1). The resulting polar plots (Fig.
9) indicate BBOA particles mainly arriving from south-easterly directions during
DIAPASON2013. During this measurement period agricultural fires were frequently observed
in the Alban Hills (Frascati vineyard area), which are located in this direction. Additionally,
BBOA was observed during conditions of north-easterly winds and higher wind speeds (up to
4 m s$^{-1}$), possibly resulting from residential wood burning in a densely populated urban
periphery area (Borghesiana). The polar plot of BBOA obtained from DIAPASON2014
measurements does hardly point to any preferential direction of BBOA origin. Since elevated
BBOA concentrations were mainly reached during low wind speed conditions, emission from
rather local sources is suggested.
BBOA was not identified in the measurements in the city centre of Rome, even not in PMF
solutions with a large number of factors (8). Since green-waste burning and domestic heating
with biomass are forbidden in central Rome, biomass burning related particles are probably
not emitted in the local environment. However, the contribution of biomass burning and
domestic heating to the urban air pollution of Rome (especially during winter time) was
reported by Gariazzo et al. (2016). Apparently, during our measurements in late spring and
autumn the contribution of particles emitted outside the suburban area is too small to be



identified with PMF and/or aging processes during the transport of the particles lead to a loss
of the BBOA fingerprint (Bougiatioti et al., 2014). Also during the MEGAPOLI
measurements in Paris emissions from biomass burning were identified in the organic aerosol
fraction (Crippa et al., 2013a). In contrast to our measurements, in Paris BBOA was only
found during the winter time campaign (Jan/Feb), but also at the measurement location in the
city centre, probably generated by local domestic wood burning (Crippa et al., 2013a).
In summary, the results from both DIAPASON measurements show that particles from
biomass burning significantly (18 – 24 % of total OA) contributed to local air pollution in the
suburban area in late autumn as well as in late spring. Agricultural fires and possibly wild
fires probably are their most important sources, since heating activities are assumed to be
quite low at these times due to the moderate temperature conditions (Table 3).

### 13    4.2.4  Emissions related to traffic and cooking activities

At both measurement locations, during both seasons particles from traffic- as well as from
cooking-related emissions were detected.
Indications of traffic-related emissions can be found in several measured variables showing a
distinct diurnal pattern with peaks during the morning and evening rush hours. The time series
and diurnal pattern of HOA (hydrocarbon-like organic aerosol), a PMF factor that is typically
associated with traffic emissions, show good agreement with the respective patterns of species
like BC, $NO_x$ and PAH (diurnal cycles: $R^2 > 0.85$) for all four campaigns. Also correlations of
complete campaign time series of HOA with BC result in good agreements ($R^2 \approx 0.7$).
In the diurnal cycles of HOA seasonal and spatial differences can be observed (Fig. 10).
Independent of season and measurement location a short peak occurs during the morning rush
hour and a broader peak starting during the evening rush hour. During all field campaigns
except DIAPASON2013, HOA concentrations remain increased throughout the night. Thus,
the exact period of the evening rush hour cannot be clearly isolated. These differences in the
shapes of the HOA peaks in the morning and evening rush hour are mainly controlled by
boundary layer dynamics together with the diurnal cycle of traffic-related emissions (rush
hour times). A seasonal difference is observed in the HOA evening rush hour peak, which
peaks around midnight during May/June, but around 7-8 pm during Oct/Nov. This shift and




the broadening of the HOA peak in May/June 2014 is probably driven by the different
boundary layer dynamics during the two seasons.
For both measurement years a time shift of the morning peak between Tor Vergata and central
Rome (later by about one hour) can be observed. Since similar diurnal temperature profiles
measured at the suburban and the urban location suggest also similar boundary layer
dynamics at the two sites, the observed shift possibly results because traffic starts in the
suburbs earlier in the morning and continues slowly towards the city centre. In contrast to our
observations, from BC measurements during the MEGAPOLI summer campaign in Paris no
distinct shift of the morning rush hour peak was observed between the two suburban and the
urban measurement locations (Freutel et al., 2013).
Mean HOA mass concentrations for the individual measurement campaigns range between
0.59 - 0.93 µg m$^{-3}$. During the 2013 measurements (Oct/Nov) similar concentrations were
obtained at the suburban site ($0.76 \pm 1.04$ µg m$^{-3}$) and central Rome ($0.71 \pm 0.72$ µg m$^{-3}$),
whereas in 2014 higher concentrations were reached at central Rome ($0.93 \pm 0.73$ µg m$^{-3}$)
compared to the suburb ($0.59 \pm 0.60$ µg m$^{-3}$). Overall, the contribution of traffic-related
emissions (e.g. HOA, NO$_x$, PAH) to local air pollutant levels was higher in central Rome, as
already discussed in Sect. 4.1.
A factor associated with cooking emissions, COA (cooking OA), was obtained by PMF
analysis of the OA measured at both locations and during both seasons. The COA mass
spectra show prominent peaks at *m/z* 41 and 55 (Allan et al., 2010; Lanz et al., 2007) and a
smaller contribution of *m/z* 60 and 73 (Mohr et al., 2009). Our COA mass spectra correlated
well with those found by Faber et al. (2013) and Mohr et al. (2012) with $R^2 = 0.63$-0.93.
The COA diurnal cycles observed at central Rome (Fig. 11, upper panel) are consistent with
results from previous studies (e.g. Allan et al., 2010; Mohr et al., 2012) showing highest
concentrations in the late evening (around 10 pm) and a smaller peak around midday (2-3
pm). This pattern is generated by a combination of source strengths and boundary layer
dynamics, with typically increased boundary layer height during lunch time compared to
dinner time.
In contrast, diurnal cycles of the COA factors measured at the suburban location in 2013 and
2014 (Fig. 11, lower panel) both show a peak in the evening, but only during
DIAPASON2014 a slight and barely significant COA concentration increase was observed
during lunch time. This could be due to an insufficient separation of the COA and HOA factor





during PMF analysis, which is also demonstrated in the COA "morning peak" of the DIAPASON2014 measurements. However, the missing midday peak also reflects the generally low abundance of cooking-related OA at the suburban measurement location: while there are strong cooking activities and a large abundance and closeness of restaurants around the central Rome site, potential sources in the immediate vicinity of the suburban site are scarce. At a distance of around 250 m from our monitoring site, a cafeteria served hot meals for lunch, but apparently, our measurements were not strongly affected by its emissions.

Consistently, absolute mass concentrations of cooking-related emissions were higher at the central Rome site ($0.70 \pm 1.00$ µg m$^{-3}$, $0.65 \pm 0.69$ µg m$^{-3}$ in 2013 and 2014, respectively) compared to the suburban measurement location ($0.45 \pm 0.50$ µg m$^{-3}$, $0.53 \pm 1.29$ µg m$^{-3}$). Ranging between 8-29 % of the total OA concentrations, cooking activities contribute significantly to (sub-) urban air pollution. During meal times the contribution of COA to total organics can be very high: For example during lunch/dinner times at central Rome, COA contribution to total organics was 35 %/53 % (POPE2013) and 9 %/25 % (POPE2014), respectively. Similar observations were made during the MEGAPOLI winter measurements in Paris, where COA contributed on average 11-17 % to total OA (up to 35 % during lunch times) (Crippa et al., 2013a).

### 4.2.5 Cigarette smoking emissions

For both POPE campaigns in central Rome PMF analysis of the organic aerosol fraction resulted in a factor which could be associated with cigarette smoke (CSOA; excluded from Fig. 2). This was not very surprising, since cigarette smoking took place in the direct vicinity of the measurement location. The mass spectra of CSOA from both years show good correlation with each other ($R^2 = 0.7$; Fig. 12). Very characteristic for the CSOA spectra is a peak at $m/z$ 84 from $C_5H_{10}N^+$ (Fig. 12). This ion ($N$-methylpyrrolidine) is typically observed in EI mass spectra of nicotine (NIST: http://webbook.nist.gov, last access 09.05.2016) and is generated by cleavage of the nicotine molecule into two heterocycles (Jacob III and Byrd, 1999). Since nicotine is one of the most abundant particulate compounds identified in cigarette smoke samples (Rogge et al., 1994), its fragments are suitable tracers for cigarette emissions. While cigarette smoke-related aerosol has been found in AMS measurements previously (Faber et al., 2013; Fröhlich et al., 2015) and also the detection of nicotine from cigarette smoke was mentioned (Jayne et al., 2000), to our knowledge, the identification of





the nicotine fragment *N*-methylpyrrolidine from analysis of HR-ToF-AMS data is reported
here for the first time. The time series of $C_5H_{10}N^+$ was used during the evaluation of the PMF
results as tracer for CSOA, yielding good correlations ($R^2 > 0.9$) with the time series of
CSOA.
The CSOA mass spectra from both POPE campaigns show reasonable to very good
agreement with CSOA mass spectra reported by Faber et al. (2013) ($0.65 < R^2 < 0.96$). Some
differences are observed between the mass spectra obtained for POPE2013 and 2014 in the
relative fraction of $CO_2^+$ and its related ions, which also affects the observed elemental ratios
(Fig. 12). This results in a potential error in CSOA concentrations of less than 10 % and
probably is due to a PMF artefact and/or insufficient correction for gas phase $CO_2$.
Comparison with mass spectra of cigarette smoke obtained in the laboratory (Faber et al.,
2013) show a contribution of $CO_2^+$ more similar to the POPE2013 measurements. Further
laboratory work in order to obtain more robust source spectra is needed to better constrain the
expected f44 in CSOA mass spectra. Due to the low intensity of the N-containing ions and a
conservative selection of ions which were fitted in the mass spectra reported by Faber et al.
(2013), the nicotine fragment ($C_5H_{10}N^+$) was not observed in their measurements of cigarette
smoke. However, after re-analysis of the mass spectra with integration of the $C_5H_{10}N^+$ ion in
the fitting procedure, a contribution of the nicotine tracer ion is clearly visible (Peter Faber,
personal communication).
Also the time series of mass concentration of the CSOA factor clearly support its attribution
to cigarette smoking emissions. The diurnal cycle of the CSOA factor strongly correlates with
typical working hours at the measurement location and with the diurnal cycle of the marker
fragment $C_5H_{10}N^+$ ($R^2 = 0.98$ for POPE2013 and POPE2014), exemplarily shown for
POPE2014 in Fig. 13 (top). Averaged CSOA mass concentrations for each day of the week
(Fig. 13; bottom) show distinct differences between working days and weekend, when the
administration of the hospital where the measurements took place was closed, supporting the
attribution of this PMF factor to locally emitted CSOA. Very similar observations were made
during the POPE2013 measurements.
Particles from cigarette smoke contributed 9-24 % (0.62-0.76 µg m$^{-3}$) to the total organic
aerosol mass measured at the location in central Rome. No indications for cigarette emissions
were found during the DIAPASON measurements. This result shows a potentially strong
influence on air quality in the direct environment of smokers, like it was also observed by
Faber et al. (2013) and Fröhlich et al. (2015). Since in our measurement, CSOA is mostly





produced close to the measurement location (i.e., can be regarded a local contamination), it
was not included in the previous analyses of organic aerosol composition (see Sect 4.1).
*$C_5H_{10}N^+$ as a potential CSOA marker ion:* The ion $C_5H_{10}N^+$ was further investigated in order
to assess its applicability and limitations as a tracer for cigarette emissions in AMS data sets.
While the nicotine fragment ion $C_5H_{10}N^+$ (*m/z* 84.08) seems to be unique to cigarette
emissions, the proximity of the ions $C_5H_8O^+$ (*m/z* 84.06) and $C_6H_{12}^+$ (*m/z* 84.09) in the mass
spectra causes interferences, since the mass resolution of the instrument (R $\approx$ 2000 in V-
mode) is not sufficient for completely separating the individual ion signals. Mass spectra of
the primary organic aerosol PMF factors (HOA, COA, BBOA) which are not related to
cigarette emissions ("$POA_{noCSOA}$") show significant contributions (0.2 to 0.7 % of total
organics mass spectral signal) of the ions $C_5H_8O^+$ and $C_6H_{12}^+$, leading to an artificial increase
of the nicotine tracer ion signal. In contrast, for OOA mass spectra the contribution of these
ions is comparatively low (0.1 % of total organics mass spectral signal), which is why a
potential interference of OOA was neglected in the following considerations.
AMS measurements performed during DIAPASON2013 and DIAPASON2014, which are
assumed to be not influenced by local cigarette emissions, were used to quantify the
concentration-dependent influence of $POA_{noCSOA}$ on the nicotine tracer ion $C_5H_{10}N^+$ due to
fitting interferences from the neighbour ions $C_5H_8O^+$ and $C_6H_{12}^+$. A linear relationship
between $C_5H_{10}N^+$ and $POA_{noCSOA}$ was observed for a $POA_{noCSOA}$ concentration range of 0-10
µg m$^{-3}$. Using the mass contribution of $C_5H_{10}N^+$ to the total CSOA mass spectra (Fig. 12;
1.8 % and 1.9 % for POPE2013/2014, respectively), the corresponding ion signals were
converted into CSOA detection limits. It was found that, under conservative considerations,
CSOA concentrations of at least 10 % of $POA_{noCSOA}$ are needed in order to exceed detection
limits. During conditions of negligible $POA_{noCSOA}$ concentrations, a CSOA detection limit of
80 ng m$^{-3}$ was estimated.
Based on these estimations, it can be concluded that $C_5H_{10}N^+$ is a suitable nicotine tracer ion
for HR-AMS measurements which are influenced by local cigarette emissions (i.e., CSOA
larger than 10 % of $POA_{noCSOA}$) and can be used to estimate CSOA concentrations or to
identify a CSOA factor from PMF analysis. Urban background concentrations of cigarette-
related particles in the range of 1 % of $PM_1$, as reported by Rogge et al. (1994), however, are
below the estimated CSOA detection limits. This is still true when considering typical
contributions of OOA to total organics (~50 %) and of OA to $PM_1$ (also ~50 %), leading to a
CSOA detection limit of around 2.5 % of $PM_1$. In order to identify cigarette smoke




contributions in the order of 1 % of $PM_1$, higher mass spectral resolution is needed to be able
to separate the nicotine tracer ion from its neighbouring ions. Nevertheless, the fitting of
$C_5H_{10}N^+$ at *m/z* 84 could be worthwhile for HR-AMS data sets which are potentially
influenced by cigarette-related particles. In future work, it should be investigated how aging
processes affect the appearance of this marker ion in the mass spectra.

## 7  5    Summary and conclusions

Intensive field campaigns have been performed during Oct/Nov 2013 and May/June 2014,
each time consecutively at two locations (suburban, urban) in the area of Rome, enabling the
study of seasonal and spatial differences of aerosol and trace gas characteristics.
During both years at the suburban location an impact of advected Saharan dust on $PM_{10}$ levels
was detected. With respect to background conditions, increases of $PM_{10}$ by 68 % (2013) and
100 % (2014) were measured, corresponding to average absolute increases of about 10 µg
$m^{-3}$. No influence of the dust occurrence on the non-refractory $PM_1$ chemical composition was
found during the advections.
At both locations, during the Oct/Nov measurements air quality was more strongly influenced
by primary emissions (e.g. BC, $NO_x$, PAH) with generally increased particle number
concentration (PNC), whereas during May/June the contribution from secondary particles
(sulphate, aged OA) and ozone was more important. Also during May/June 2014, new particle
formation was frequently detected around midday, while in the colder season no distinct
increase of PNC took place outside typical rush hour times. The consequence of higher
temperatures plus stronger solar radiation was also visible in the SOA-related fraction of the
organic aerosol: During the warmer season two types of OOA (less oxidised, fresher SV-
OOA and strongly oxidised, older LV-OOA) were identified, while during the colder season
only strongly oxidised OOA was found.
Typical tracers for anthropogenic emissions ($CO_2$, $NO_x$, PAH, HOA, COA) were increased at
the urban measurement location. However, absolute concentrations of POA were higher at the
suburban location, due to a strong contribution from biomass burning OA, which here
accounted for 1.28 µg $m^{-3}$ (24 % of total OA) and for 0.82 µg $m^{-3}$ (18 % of total OA) in
Oct/Nov 2013 and May/June 2014, respectively. To a large degree this was related to
agricultural waste burning in the surrounding areas and during Oct/Nov2013 potentially also
to residential wood burning in the urban periphery.





Cooking- and traffic-related aerosol was observed at both locations during both seasons. The
diurnal cycles of HOA (traffic-related OA) always peaked during rush hour times. A time
shift in the morning rush hour peak between the suburban site and central Rome was
observed, likely as a consequence of traffic progressing from the suburbs to the city centre.
HOA accounted for 0.59 to 0.93 µg m$^{-3}$ (13 to 29 % of OA) at the different locations and
seasons. COA, as an indicator for cooking activities, showed maximum concentrations during
lunch and dinner times at central Rome, whereas at the suburban location only during dinner
times distinct peaks were observed. Average COA mass concentrations of 0.45 to 0.70 µg m$^{-3}$
(8 to 29 % of OA) were found, with higher concentrations observed at central Rome
compared to the suburban location (2013: +50 %; 2014: +23 %), as expected due to the higher
density of related sources.
A type of OA related to nearby cigarette emissions (CSOA) was detected at central Rome,
and found to strongly correlate with a characteristic nicotine fragment (*N*-methylpyrrolidine,
$C_5H_{10}N^+$) at *m/z* 84 in the mass spectra. This ion could serve as a suitable tracer for locally
emitted cigarette smoke also for other datasets. However, in order to identify CSOA based
solely on this tracer ion, CSOA must account for at least 10 % of the sum of COA, HOA and
BBOA, due to interferences of neighbouring ion signals from these POA types. In the absence
of those, a detection limit of 80 ng m$^{-3}$ was found for CSOA. These findings imply that the
resolution of the HR-AMS is not sufficient to identify urban background contributions of
cigarette emissions (~1 % of PM$_1$, Rogge et al., 1994) based solely on $C_5H_{10}N^+$, while fitting
of this ion could be worthwhile for HR-AMS datasets which are potentially influenced by
nearby cigarette emissions.
During our measurements sub-micron aerosol originating from sources in the metropolitan
area of Rome and particles being advected from outside (dust periods were excluded)
contributed 42-70 % and 30-58 % to total measured PM$_1$, respectively. Thus, during our
measurements approximately half of the locally measured PM$_1$ was "home-made".
While for individual aerosol types clear spatial and temporal characteristics were observed
and can be understood, no general conclusion can be drawn whether total aerosol mass
concentrations are generally higher at the suburb or the city centre. Instead, consistent with
observations made in the area of Paris (Freutel et al., 2013), it was found that aerosol levels
strongly depend on the combination of meteorological conditions (e.g. origin of air masses,
dilution capacity within the boundary layer) and contributions of secondary aerosols and local
emissions.



## 1  Acknowledgements

The authors thank the DIAPASON team, Thomas Böttger, Antonios Dragoneas and the hospital Santo Spirito in Sassia for logistical and technical support during the field campaigns. We also thank the Arpa Lazio Environmental Agency for providing TKE data, the Barcelona Supercomputing Center for providing the BSC-DREAM8b model, and NOAA Air Resources Laboratory (ARL) for the provision of the HYSPLIT transport and dispersion model. Franziska Köllner and Daniel Kunkel are gratefully acknowledged for providing the FLEXPART data.

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



1    Table 1. Summary of measurement campaigns and measurement periods.

| Campaign name | Measurement location | Classification | Measurement period |
|---|---|---|---|
| DIAPASON2013 | Tor Vergata | Urban background | 23.10. – 07.11.2013 |
| POPE2013 | Central Rome | Urban | 07.11. – 14.11.2013 |
| DIAPASON2014 | Tor Vergata | Urban background | 20.05. – 04.06.2014 |
| POPE2014 | Central Rome | Urban | 04.06. – 17.06.2014 |



Table 2. Summary of the instruments deployed in MoLa during all measurement periods,
together with measured variables. $D_p$: particle diameter (defined according to individual
instrumental measurement method: optical, aerodynamic, or mobility diameter).

| Instrument | Measured variable |
| --- | --- |
| **Chemical composition of PM$_1$** | |
| Aerosol mass spectrometer (HR-ToF-AMS) | Organics, sulphate, nitrate, ammonium, chloride (non-refractory) mass concentrations ($D_p = \sim 70 - 800$ nm) |
| Multi Angle Absorption Photometer (MAAP) | Black carbon (BC) mass concentration |
| PAS2000 | Particulate PAH[a] mass concentration |
| **Physical aerosol properties** | |
| Condensation particle counter (CPC) | Total number concentration ($D_p > 2.5$ nm) |
| Fast mobility particle sizer (FMPS) | Size distribution ($D_p = 5.6 - 560$ nm) |
| Optical particle counter (OPC)[b] | Size distribution ($D_p = 0.25 - 32$ µm) |
| Aerodynamic particle sizer (APS) | Size distribution ($D_p = 0.5 - 20$ µm) |
| Environmental Dust Monitor (EDM) | Mass concentration of PM$_1$, PM$_{2.5}$, PM$_{10}$ |
| **Trace gas mixing ratios** | |
| Airpointer | NO$_2$, NO$_x$, NO, SO$_2$, CO, O$_3$ |
| LI-840 | H$_2$O, CO$_2$ |
| **Meteorology** | |
| Meteorological station[c] | Wind direction, wind speed, temperature, pressure, solar radiation, precipitation, relative humidity |

[a] Polycyclic Aromatic Hydrocarbons
[b] No data collected during DIAPASON2014.
[c] Wind direction, wind speed and solar radiation data not useable during POPE measurements.



1    Table 3. Summary of selected variables measured during DIAPASON (Tor Vergata) and

2    POPE (central Rome) in 2013 and 2014. Values represent total campaign averages, calculated

3    from 5 min averages, and their standard deviations. N/A: not available.

| | DIAPASON2013 (Oct/Nov 2013) | POPE2013 (Nov 2013) | DIAPASON2014 (May/June 2014) | POPE2014 (June 2014) |
|---|---|---|---|---|
| Temperature / °C | $17.6 \pm 2.9$ | $16.1 \pm 2.8$ | $19.0 \pm 3.5$ | $24.9 \pm 4.5$ |
| Rain [a] / mm | 46.3 (5) | 25.9 (6) | 6.3 (5) | 69.9 (4) |
| Pressure / hPa | $1004 \pm 7$ | $1009 \pm 6$ | $1002 \pm 2$ | $1012 \pm 4$ |
| Solar radiation [b] / W m$^{-2}$ | $103 \pm 26$ | N/A | $282 \pm 55$ | N/A |
| Rel. humidity / % | $78 \pm 11$ | $73 \pm 6$ | $61 \pm 15$ | $52 \pm 16$ |
| Wind speed / m s$^{-1}$ | $1.9 \pm 1.3$ | N/A | $2.5 \pm 1.6$ | N/A |
| Daily BLH [c] max. / m | $850 \pm 220$ | $900 \pm 150$ | $1500 \pm 450$ | $1560 \pm 250$ |
| TKE [d] / J kg$^{-1}$ | $0.48 \pm 0.45$ | $1.8 \pm 2.3$ | $0.84 \pm 0.79$ | $0.68 \pm 0.52$ |
| $PM_{10}$ / µg m$^{-3}$ | $22 \pm 12$ | $13 \pm 7$ | $15 \pm 8$ | $17 \pm 8$ |
| $PM_{10-2.5}$ [e] / µg m$^{-3}$ | $4.5 \pm 3.1$ | $4.7 \pm 3.2$ | $9.3 \pm 6.5$ | $7.1 \pm 5.7$ |
| $PM_1$ / µg m$^{-3}$ | $15 \pm 10$ | $6.0 \pm 3.6$ | $5.8 \pm 3.5$ | $7.5 \pm 3.8$ |
| PNC / $10^3$ cm$^{-3}$ | $23 \pm 15$ | $27 \pm 11$ | $18 \pm 10$ | $13 \pm 5$ |
| $NO_x$ / ppb | $29 \pm 27$ | $36 \pm 27$ | $9 \pm 9$ | $13 \pm 8$ |
| $CO_2$ / ppm | $410 \pm 20$ | $420 \pm 20$ | $410 \pm 20$ | $420 \pm 20$ |
| $O_3$ / ppb | $14 \pm 14$ | $8.9 \pm 10$ | $34 \pm 19$ | $35 \pm 22$ |
| PAH / ng m$^{-3}$ | $45 \pm 54$ | $45 \pm 39$ | $10 \pm 14$ | $12 \pm 11$ |
| Org [f] / µg m$^{-3}$ | $5.3 \pm 4.5$ | $2.5 \pm 1.8$ | $4.5 \pm 3.2$ | $6.6 \pm 3.3$ |
| $SO_4$ / µg m$^{-3}$ | $2.0 \pm 1.1$ | $0.48 \pm 0.44$ | $1.6 \pm 0.44$ | $2.6 \pm 1.3$ |



| | | | | |
|---|---|---|---|---|
| $NO_3$ / µg m$^{-3}$ | $0.86 \pm 0.80$ | $0.23 \pm 0.17$ | $0.61 \pm 0.72$ | $0.49 \pm 0.40$ |
| $NH_4$ / µg m$^{-3}$ | $0.88 \pm 0.48$ | $0.20 \pm 0.18$ | $0.66 \pm 0.27$ | $0.97 \pm 0.46$ |
| Chl / µg m$^{-3}$ | $0.09 \pm 0.16$ | $0.04 \pm 0.06$ | $0.06 \pm 0.13$ | $0.03 \pm 0.08$ |
| BC / µg m$^{-3}$ | $2.9 \pm 2.5$ | $2.2 \pm 1.7$ | $1.3 \pm 1.0$ | $1.8 \pm 1.0$ |

[a] Total accumulated amount of rain during measurements. Numbers of days with rain are given in parentheses.
[b] Average and standard deviation of daily means. Only days with 24 h measurements were used (includes 85 % of data).
[c] Average and standard deviation of daily boundary layer height (BLH) maxima from polarization LIDAR-ceilometer.
measurements at the Tor Vergata site provided by the DIAPASON project (DIAPASON, 2016).
[d] Turbulent kinetic energy (TKE) calculated from 2-hour averages provided by Arpa Lazio Environmental Agency from
measurements at 3 sites around Rome (Tor Vergata, Castel di Guido, Boncompagni).
[e] Difference between $PM_{10}$ and $PM_{2.5}$ (coarse particles).
[f] For POPE2013 and POPE2014 corrected for contribution from local cigarette smoke emissions, compare Sect. 4.2.5.



1    Table 4: Estimated contribution of "home-made" and "advected" species to total $PM_1$ for all

2    measurement periods. Dust advection periods and emissions from cigarettes are excluded.

| | Home-made $PM_1$ / µg m$^{-3}$ (contribution to $PM_1$ / %) | Advected $PM_1$ / µg m$^{-3}$ (contribution to $PM_1$ / %) |
|---|---|---|
| DIAPASON2013 | 6.5 (47) | 7.3 (53) |
| POPE2013 | 3.9 (70) | 1.7 (30) |
| DIAPASON2014 | 4.7 (59) | 3.3 (41) |
| POPE2014 | 5.2 (42) | 7.3 (58) |



1    Table 5. Summary of "dust" and "no dust" periods identified during DIAPASON2013 and

2    DIAPASON2014, including mean values and standard deviation of $PM_{10-2.5}$ and $PM_{10}$.

| | DIAPASON2013 | | DIAPASON2014 | |
|---|---|---|---|---|
| | Dust | No dust | Dust | No dust |
| Period | 29.10.-01.11.13 | 23.10.-28.10.13; 02.11.-07.11.13 | 20.05.-26.05.14 | 27.05.-04.06.14 |
| Mean $PM_{10-2.5}$ / $\mu g\ m^{-3}$ | $7.1 \pm 3.5$ | $4.0 \pm 2.7$ | $15.4 \pm 5.9$ | $5.5 \pm 2.8$ |
| Mean $PM_{10}$ / $\mu g\ m^{-3}$ | $32 \pm 7$ | $19 \pm 12$ | $22 \pm 7$ | $11 \pm 5$ |
| Fraction of $PM_{10}$ in $PM_1$ | 70 % | 68 % | 31 % | 48 % |





Table 6. Variables measured during DIAPASON2014 indicating different ambient conditions on days classified as "nucleation", "non-defined" and "non-nucleation" days. Total particle number concentrations (PNC) and the classification criteria and number of respectively classified days are also listed. For each measurement day and presented variable daily averages for the period 10 am – 4 pm were calculated. Here, mean and standard deviation of these averages are shown with maximum values coloured in red, minimum values coloured in green.

| | Nucleation days | Non-defined days | Non-nucleation days |
|---|---|---|---|
| Classification | $PNC_{nuc}/PNC_{bg} \geq 1.5$ | $1 < PNC_{nuc}/PNC_{bg} < 1.5$ | $PNC_{nuc}/PNC_{bg} \leq 1$ |
| Number of days | 6 | 6 | 2 |
| Rain [a] / mm | 0 (0) | 0.7 (1) | 2.3 (1) |
| Total PNC / $10^3$ cm$^{-3}$ | $25 \pm 9$ | $13 \pm 3$ | $12 \pm 1$ |
| Temperature / ° C | $23 \pm 2$ | $22 \pm 2$ | $20 \pm 0$ |
| Rel. humidity / % | $43 \pm 6$ | $46 \pm 9$ | $56 \pm 4$ |
| Solar radiation / W m$^{-2}$ | $780 \pm 80$ | $700 \pm 190$ | $670 \pm 200$ |
| $O_3$ / ppb | $54 \pm 4$ | $47 \pm 6$ | $48 \pm 1$ |
| Total $PM_1$ organics / µg m$^{-3}$ | $3.9 \pm 2.4$ | $3.6 \pm 1.0$ | $2.9 \pm 2.3$ |

[a] Total accumulated amount of rain (10 am – 4 pm). Numbers of days with rain are given in parentheses.





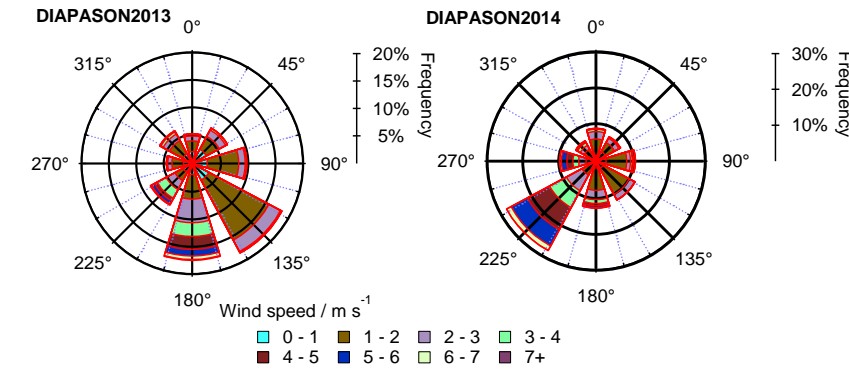

2    Figure 1. Relative frequency of local wind directions (in °) colour coded with wind speed

3    measured during DIAPASON2013 (left) and DIAPASON2014 (right).





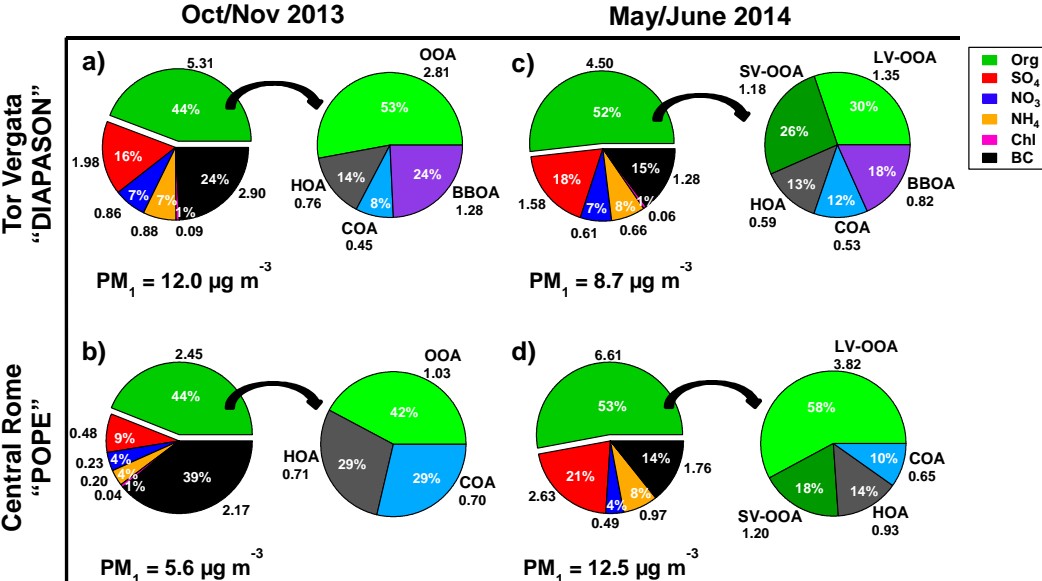

Figure 2. Mean chemical composition (µg m$^{-3}$) of non-refractory PM$_1$ together with BC (left chart in panels a-d) and PMF-separated organic fraction (right chart in panels a-d) for each measurement period. PM$_1$ values below the pie charts represent total mass concentration of AMS-measured species plus BC. The organic fraction measured at central Rome was corrected for contributions from cigarette smoke in the local environment (Sect. 4.2.5, also omitted from the pie charts depicting the PMF-separated organic fraction).




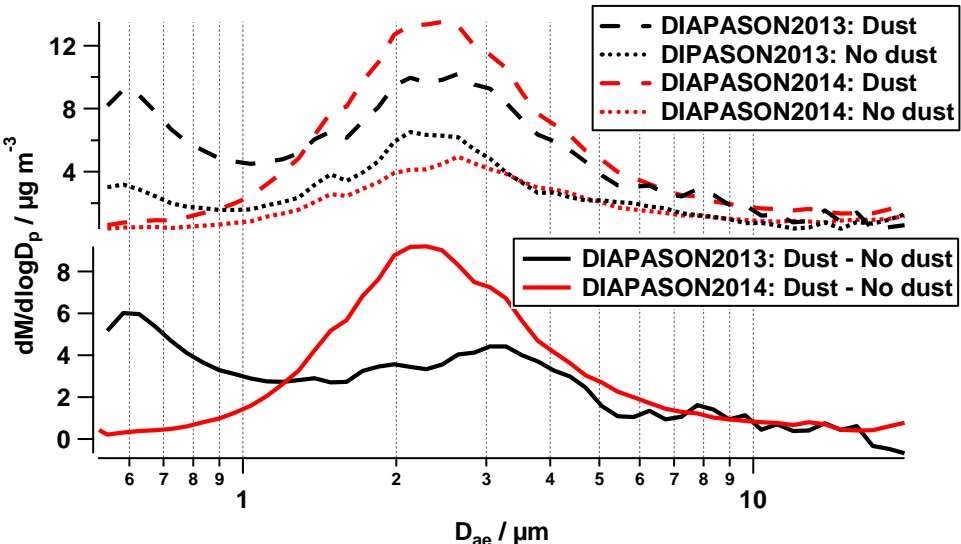

Figure 3. Size-resolved mass distribution (dM/dlogD$_p$) during "dust" (dashed traces) and "no

dust" (dotted traces) periods (top panel) measured with the APS during DIAPASON2013

(black traces) and DIAPASON2014 (red traces). The difference of the size-resolved mass

distributions measured during "dust" and "no dust" periods indicate the size distributions of

the dust particles measured during both years (bottom panel). D$_{ae}$ is the aerodynamic particle

diameter.





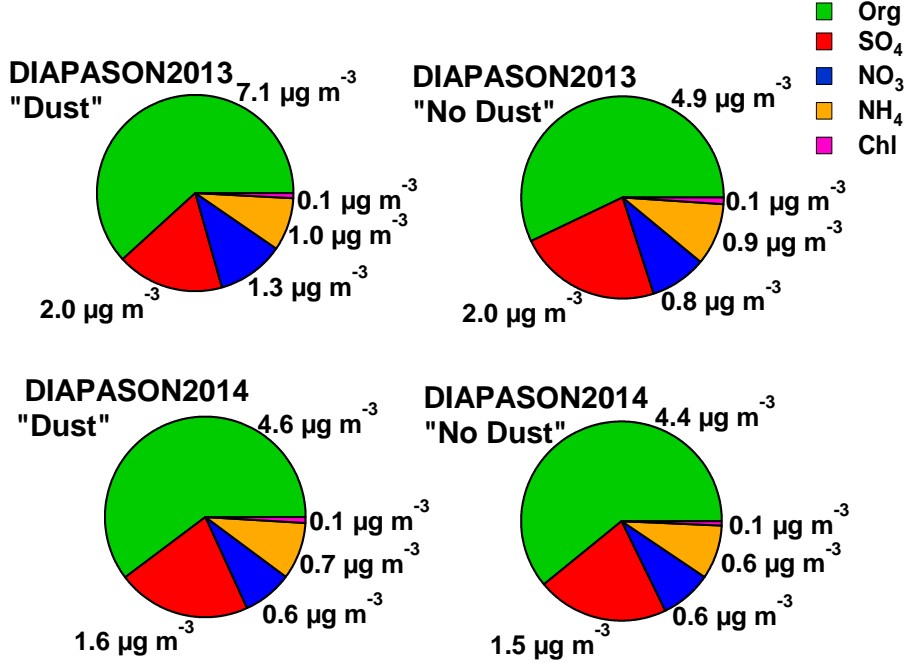

2    Figure 4. Comparison of mean chemical composition of non-refractory PM$_1$ obtained from

3    AMS measurements during "dust" (left) and "no dust" (right) periods during

4    DIAPASON2013 (top) and DIAPASON2014 (bottom).





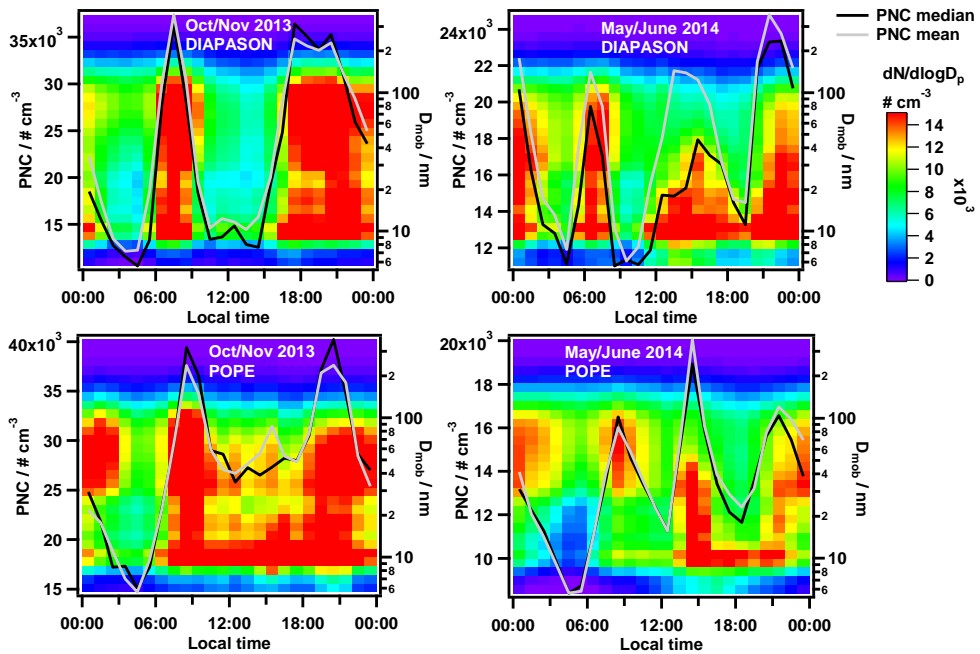

2 Figure 5. Average diurnal cycles of particle number concentrations and size distributions for

3 DIAPASON (top) and POPE (bottom) for each year (left: 2013, right: 2014). Image plots of

4 diurnal cycles of the particle number size distributions (colour coded for $dN/dlogD_p$) are

5 shown with the particle diameters on the right axes (mobility particle diameter $D_{mob}$). Mean

6 (grey) and median (black) diurnal cycles of the total particle number concentrations are shown

7 on the left axes. New particle formation at midday was only observed in the May/June 2014

8 campaigns.





Figure 6. Average size distributions of particle number concentrations (dN/dlogD$_p$) for
maximum PNC at midday (DIAPASON2014: 1-3 pm; POPE2014: 2-3 pm; solid traces) and
during background conditions (10-11 am; dashed traces) from FMPS measurements for
DIAPASON2014 (blue) and POPE2014 (red). D$_{mob}$ is the mobility diameter.





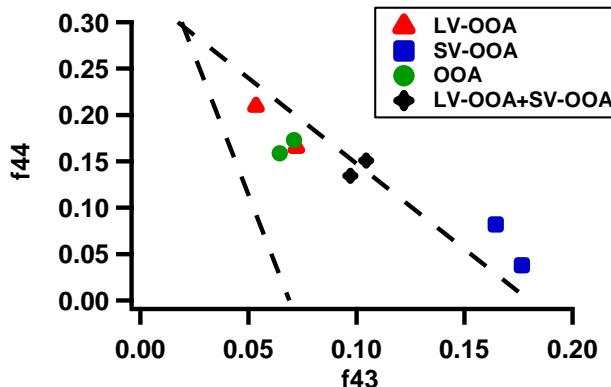

Figure 7. f44 vs. f43 plotted for each OOA factor obtained from PMF analysis of the organic
fraction of HR-AMS data. OOA factors (green markers) resulted from DIAPASON2013 and
POPE2013 measurements; SV-OOA (blue markers) and LV-OOA factors (red markers) were
found during DIAPASON2014 and POPE2014 measurements. The recombination of the
factors LV-OOA and SV-OOA for both DIAPASON2014 and POPE2014 is also shown
(black markers). The dashed lines represent the triangular space in which measured ambient
OOA components typically cluster according to Ng et al. (2010).





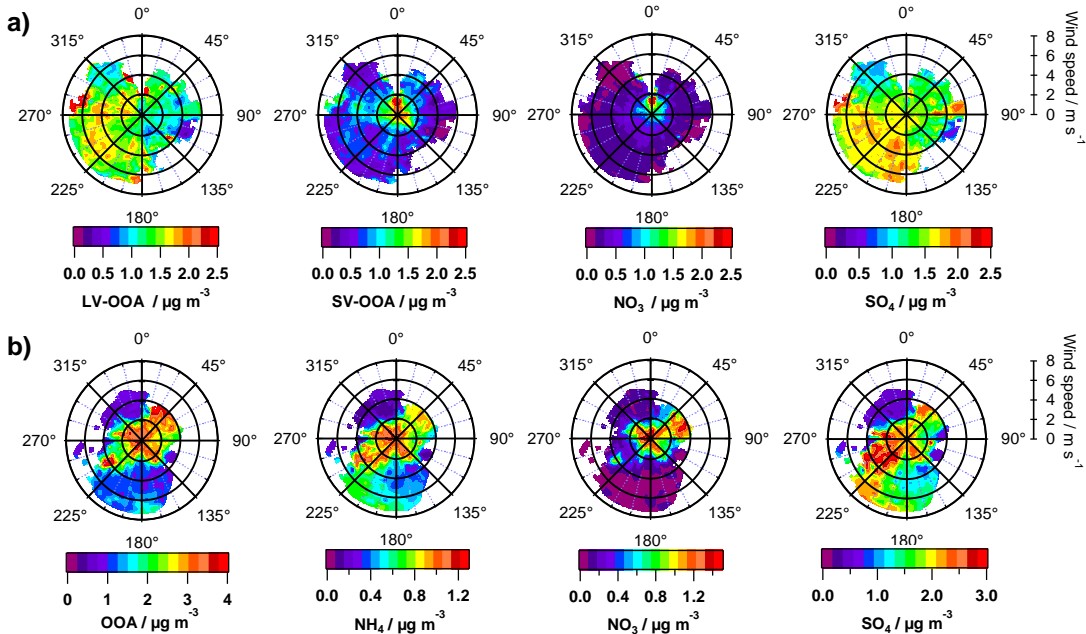

Figure 8. Polar plots of (a) submicron LV-OOA, SV-OOA, $NO_3$ and $SO_4$ concentration

(colour coded) obtained from DIAPASON2014 measurements and (b) submicron OOA, $NH_4$,

$NO_3$ and $SO_4$ concentration obtained from DIAPASON2013 measurements as a function of

local wind direction (°) and speed (m s$^{-1}$).





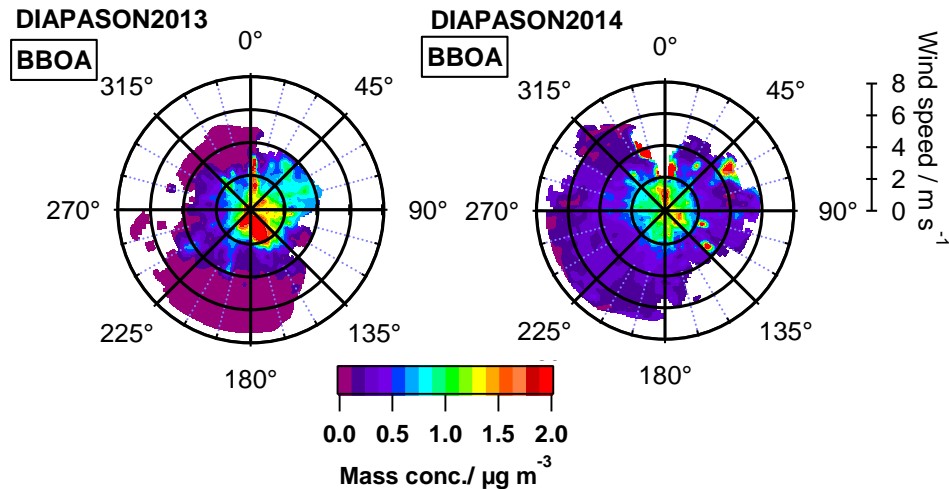

2    Figure 9. Submicron BBOA mass concentrations (colour coded) as a function of local wind

3    direction (°) and speed (m s$^{-1}$) for DIAPASON2013 (left) and DIAPASON2014 (right).





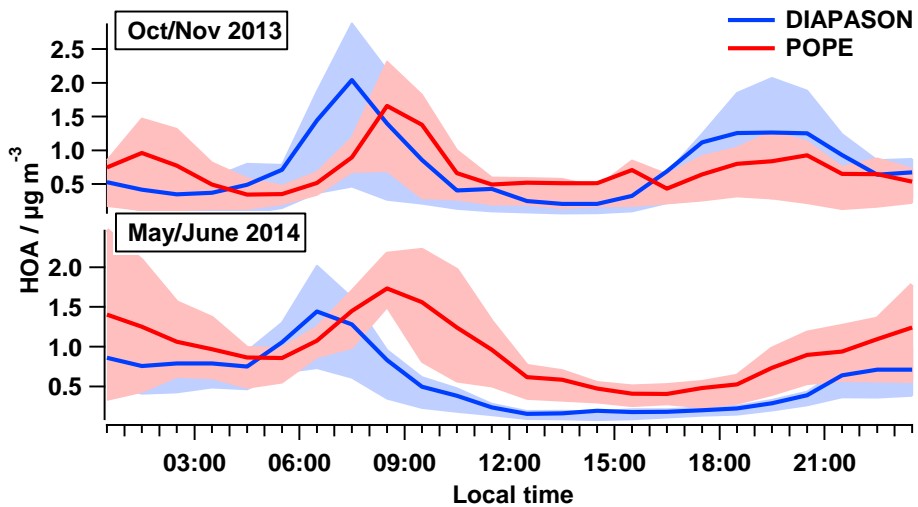

2      Figure 10. Diurnal cycles of HOA mass concentrations obtained from measurements at Tor

3      Vergata (blue; DIAPASON) and central Rome (red; POPE) during both seasons. Shown are

4      mean concentrations (traces) and the corresponding 25th and 75th percentiles (shaded areas).

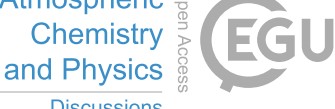



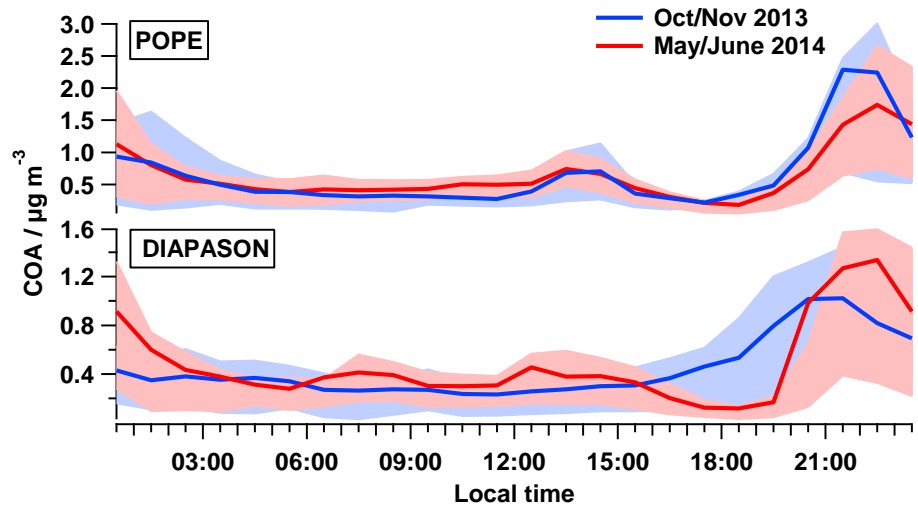

2 Figure 11. Diurnal cycles of COA mass concentrations observed at central Rome (top; POPE)

3 and the suburban site (bottom; DIAPASON) during both seasons. Shown are mean

4 concentrations (traces) and the corresponding 25th and 75th percentiles (shaded areas).





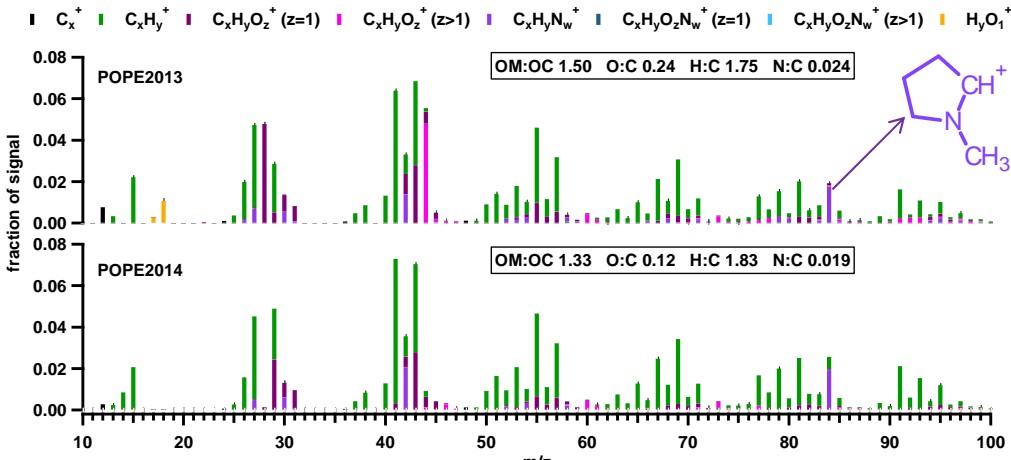

Figure 12. Unit mass resolution spectra of CSOA obtained for the two POPE campaigns,
calculated from organic high resolution mass spectra and colour coded for the different groups
of ion fragments. The elemental ratios are shown in boxes. The chemical structure of the
suggested ion fragment at $m/z$ 84 ($C_5H_{10}N^+$) is also illustrated.





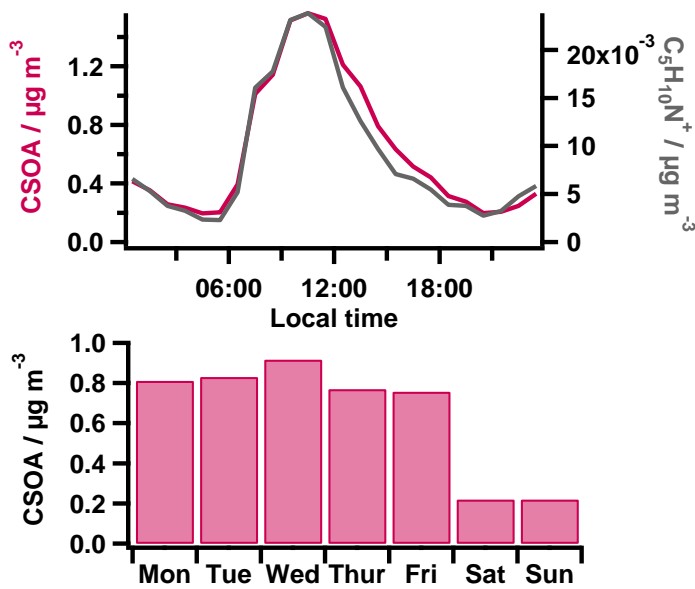

2    Figure 13. Diurnal cycle of CSOA and the marker fragment $C_5H_{10}N^+$ (top) and the weekly

3    cycle (bottom) of CSOA mass concentrations obtained from HR-AMS data of POPE2014.