# Peer review of "Atmospheric aerosols in Rome, Italy: Sources, dynamics"

_Atmospheric Chemistry and Physics, 2016_

## Referee Comment (RC1) · Anonymous Referee #1 · 18 Aug 2016

The manuscript by Struckmeier et al. analyzed four datasets that were collected at two sites in different seasons in Rome (suburban vs. urban). This study contains various real-time online measurements including aerosol chemical composition, gaseous species, particle number concentrations, and meteorological parameters. The sources of organic aerosols (OA) were also analyzed with positive matrix factorization. While rich chemical information was provided in this work to address the sources, dynamics, and spatial variations, the discussions e.g., composition, dust event, new particle formation, OA, and CSOA are scattered and lack of focus. Also, I have several major concerns on data analysis and the interpretations: (1) each campaign lasted less than two weeks, and most importantly, the measurements at the suburban and urban

sites were not simultaneous. This clearly increases the uncertainties in comparing aerosol chemistry and sources between the two sites. In addition, it is difficult to see the dynamic variations of aerosol species in Rome if the authors didn't present time series data. (2) the data quality was not validated adequately, particularly the AMS measurements. A simple comparison between PM1 measured by EDM and that measured by AMS and MAAP (NR-PM1 + BC) will help. (3) the AMS data analysis needed to be expanded. For example, which approach (Aiken et al., 2008 or Canagaratna et al., 2015) was used to calculate the elemental ratios the calculation of elemental ratios? If there are elemental ratios, why did the authors still use f43 and f44 to discuss the oxidation states? (4) the PMF analysis is a big weakness of this study. The authors didn't have a full evaluation of the PMF results. At least, the authors need to present the mass spectral profiles and times series of all OA factors, and also the comparisons with collocated measurements. The diurnal correlations the authors mentioned in page 21 did not mean much. Figure 2 also showed substantial differences in HOA/BC ratios at the two sites in different seasons, and surprising BC contributions, which should be well interpreted. The PMF uncertainties lead to another major concern of the cigarette smoking factor. Although the authors concluded this as a major finding and presented a long discussion on it, it is still not convincing due to the limited resolution of V-mode (C5H10N+) and the absence of the measurements of molecular markers for cigarette smoking. I am also suspicious that the diurnal profile of CSOA did not reflect cigarette smoking that is expected less affected by boundary layer dynamics (if the authors claimed it as a point source). Showing the times series of CSOA factor will help. (5) the new particle formation in this study appears to have problems too. At least from the average diurnal cycles in Figure 5, we didn't see "banana" shape. On the other hand, the diurnal cycles appears to indicate strong local sources at both sites. (6) the classifications of "home-made" and "advected" might also have large uncertainties. For example, OOA can be from both sources since SV-OOA and LV-OOA cannot be separated. Although nitrate has a shorter life time than sulfate and LV-OOA, many studies have shown that regional transport can be important. I understand the

authors can judge this based on the polar plots in Figure 8. In fact, I suggest that the authors re-analyze the polar plots by considering the influences of the number points in each cell. For example, the wind rose plots in Figure 1b shows a small frequency from the northeast, the polar plots in this direction can be significantly biased by sporadic spikes. With this, I cannot recommend it for publication on ACP.

---

## Author Comment (AC1) · 25 Aug 2016

**Replies to comments by anonymous referee #1**

We thank referee #1 for his/her comments. We completely agree with the general attitude behind these comments that complex measurements and data processing need substantial data validation and quality assurance efforts in order to produce robust and reliable results. As detailed below, we performed a large range of data validation efforts during the processing of our measurement data. Much of this is described in the submitted manuscript; detailed information on the other efforts could have been included as well. However, to avoid an overly lengthy manuscript, we only presented statistical summaries of many validation results. If the editor feels that this would improve the paper, we are more than happy to provide a supplement with graphical information and detailed description on the data validation tests.

Comment: The manuscript by Struckmeier et al. analyzed four datasets that were collected at two sites in different seasons in Rome (suburban vs. urban). This study contains various real-time online measurements including aerosol chemical composition, gaseous species, particle number concentrations, and meteorological parameters. The sources of organic aerosols (OA) were also analyzed with positive matrix factorization. While rich chemical information was provided in this work to address the sources, dynamics, and spatial variations, the discussions e.g., composition, dust event, new particle formation, OA, and CSOA are scattered and lack of focus.

Reply:

The intention and focus of our manuscript is to provide an assessment of the sub-micron aerosol and its potential sources at an urban and a suburban location in the Rome area during different measurement periods. This includes a general overview (the "rich chemical information" mentioned by the reviewer), but also deeper discussion on specific areas, which need to be addressed in order to characterize a complex urban environment like Rome. In particular with respect to the very different sources it would have been incomplete if for example the Saharan dust or the NPF, or other aspects had been left out. Thus, the focus of our work is on the description and, if possible, explanation of aerosols from the various sources, similar to approaches in previous publications from different groups (e.g., Kostenidou et al., 2015;Crippa et al., 2013b) which show that there is a general interest in this kind of analysis.

Comment: Also, I have several major concerns on data analysis and the interpretations: (1) each campaign lasted less than two weeks, and most importantly, the measurements at the suburban and urban sites were not simultaneous. This clearly increases the uncertainties in comparing aerosol chemistry and sources between the two sites. In addition, it is difficult to see the dynamic variations of aerosol species in Rome if the authors didn't present time series data.

Reply:

The durations of the data acquisition periods are in detail:

DIAPASON2013: 16 days of measurements (including 14 full days)
POPE2013: 8 days of measurements (including 6 full days)
DIAPASON2014: 16 days of measurements (including 14 full days)
POPE2014: 14 days of measurements (including 12 full days)

This shows that only POPE2013 was shorter than two weeks. The measurements were performed subsequently and not simultaneously as stressed in the manuscript, e.g., in the first sentence of the abstract. Obviously this results in a larger uncertainty when comparing the observations from the urban and suburban site, as the reviewer correctly states. This is why we do not generalise observations or differences in observations at different locations or during different seasons, but rather present our findings considering potentially different weather conditions etc., as clearly indicated several times in the manuscript, e.g.:

- Page 12, line 5: "Regarding absolute $PM_1$ concentrations […] neither any general conclusion whether aerosol mass concentrations are higher at the city centre or in the suburb, nor whether $PM_1$ concentrations are elevated during any of the two different seasons, can be drawn."
- Page 12, line 12: "As discussed above, changes in meteorological conditions are likely one explanation for this result …"
- Page 12, line 18: "In addition to meteorological conditions (e.g. solar radiation, BLH, TKE, air mass origin, etc.) local air quality can be strongly influenced by local emission from various sources (traffic, cooking, biomass burning)."

Nevertheless, each of the four measurement intervals provides valuable information on properties and dynamics of various aerosol types in the city centre and at the suburban location, which are found to be sufficiently robust within the available measurement time.

Of course, we could provide time series of all measured parameters for each campaign. However, we decided not to focus on detailed temporal evolution or individual events but on typical aerosol properties and dynamics (e.g., diurnal cycles). In our case, where measurements were not performed simultaneously at both locations, this seems more appropriate and meaningful.

Comment: (2) the data quality was not validated adequately, particularly the AMS measurements. A simple comparison between PM1 measured by EDM and that measured by AMS and MAAP (NR-PM1 + BC) will help.

Reply:

We agree that assuring data quality is good practice. During the analysis of the AMS (and other instrument's) data all standard procedures, checks, calibrations, corrections, intercomparisons, etc. have been performed (see statements in Sections 3.1 and 3.2). Besides other necessary quality checks, also comparison of AMS plus BC data with independent $PM_1$ data (EDM, but also $PM_1$ calculated from size distribution measurements) was performed as well as many other comparisons. All comparisons showed reasonable

agreement (e.g., for $PM_1$ from AMS+BC versus EDM, $R^2$=0.69-0.85, Slopes: 0.73-1.37). We will add this kind of detailed information in our revised manuscript.

Comment: (3) the AMS data analysis needed to be expanded. For example, which approach (Aiken et al., 2008 or Canagaratna et al., 2015) was used to calculate the elemental ratios the calculation of elemental ratios? If there are elemental ratios, why did the authors still use f43 and f44 to discuss the oxidation states?

Reply:

The results from AMS measurements were inspected for relationships among these data and relationships with data from other instruments. For the data obtained with the AMS and the other instruments temporal evolution, diurnal patterns and individual events were analysed (see Section 4). The relevant results of the various quality assurance tests can be provided in a supplement to our revised manuscript.

Elemental ratios were calculated based on the current state of the art method (Canagaratna et al., 2015). We thank the reviewer for pointing out this missing information, and will add it to the revised manuscript.

Our discussion using f43 and f44 focused on the aerosol aging levels, for which the "triangle" introduced by Ng et al. (2010) is commonly used (e.g., Zhang et al., 2015;Ortega et al., 2016;Xu et al., 2016). We also calculated elemental ratios for the various factors, but didn't include them in the discussion on aerosol aging levels since they show equivalent behaviour as the f44/f43 data. We can add this information to Figure 7.

Comment: (4) the PMF analysis is a big weakness of this study. The authors didn't have a full evaluation of the PMF results. At least, the authors need to present the mass spectral profiles and times series of all OA factors, and also the comparisons with collocated measurements.

Reply:

The performed PMF analysis and corresponding tests are subject of Section 3.2. All typical and many other tests to validate the results were applied, amongst others those according to the guidelines by Zhang et al. (2011), Table 1. All relevant information (e.g. m/z-range used for PMF, method of error and data matrix preparation, treatment of isotopes, treatment of low S/N data, treatment of $CO_2$-related ions, range of factors, fPeak and seed investigated) is detailed in the text.

Furthermore, each PMF factor was inspected for reasonability and validated by comparing factor time series with external species and mass spectral profiles with such from literature. Details are given in the text, e.g., page 8, lines 17-19; page 19, lines 13-17; page 20, lines 1-7; page 21, lines 17-21; page 22, lines 19-22; page 24, lines 5-12.
So far we did not present all individual correlation plots and mass spectra in graphs, but presented this information in the form of $R^2$ values to save journal space and keep the length of the manuscript at a reasonable level without losing information. We could include this information in a supplement.

Comment: The diurnal correlations the authors mentioned in page 21 did not mean much.

Reply:

Such correlations are commonly used to show potential relationships between variables (e.g., Sun et al., 2016;Zhang et al., 2014), and to validate PMF factors (Zhang et al., 2011). In page 21 we discuss that the diurnal patterns of HOA correlate well with those of other typically traffic-related species (BC, NOx, PAH). As stated in page 21, line 21, not only the diurnal cycles, but also the time series correlate well.

Comment: Figure 2 also showed substantial differences in HOA/BC ratios at the two sites in different seasons, and surprising BC contributions, which should be well interpreted.

Reply:

We thank the reviewer for this valuable comment, and will add some more discussion on this topic. We agree that there are differences in the HOA/BC ratios observed during the fall 2013 campaigns and during the spring 2014 campaigns (fall 2013: 0.26-0.33; spring 2014: 0.46-0.53). We attribute this to the fact that both, biomass burning and traffic contribute to total BC, as mentioned on page 10, lines 20-21. Biomass burning is more predominant in fall, leading to higher BC mass concentrations, and therefore lower HOA/BC ratios during this season. The fact that larger ratios were found in the city centre in both seasons reflects the lower contribution of biomass burning emissions at this site. The contribution of different sources (traffic and biomass burning) also explains why the ratio of HOA/BC from our study is different to such ratios found in source measurements of pure traffic-related emissions. Separation of BC related to the two sources unfortunately is not possible with the instrument used (MAAP), but measurements in other cities with the aethalometer instrument (e.g., Crippa et al., 2013a) have shown that BC contributions from biomass burning to total BC can be non-negligible. We will include this discussion in the revised manuscript.

Comment: The PMF uncertainties lead to another major concern of the cigarette smoking factor. Although the authors concluded this as a major finding and presented a long discussion on it, it is still not convincing due to the limited resolution of V-mode (C5H10N+) and the absence of the measurements of molecular markers for cigarette smoking. I am also suspicious that the diurnal profile of CSOA did not reflect cigarette smoking that is expected less affected by boundary layer dynamics (if the authors claimed it as a point source). Showing the times series of CSOA factor will help.

Reply:

PMF analysis is always associated with uncertainties. As described in the manuscript we have taken all care in order to minimize these uncertainties and we have made multiple tests and comparisons with other data (e.g. mass spectra of previously identified CSOA, as described in the manuscript) in order to obtain results as robust as possible. While the CSOA factor was first obtained from the PMF analysis of the whole mass spectra, afterwards the $C_5H_{10}N^+$ marker was identified and found to have time series that correlate very well with those of

CSOA. Indeed, as discussed in the manuscript, CSOA has been identified before from AMS data in which the newly found marker fragment at m/z 84 was not fitted at all (Faber et al., 2013) and yielded a very similar factor mass spectrum with correlation coefficients $0.65 < R^2 < 0.96$ (page 24, line 6). (Re-analysis later showed that the marker fragment indeed was present also in these data, as discussed in the manuscript.) Therefore, and from other tests we performed in the data evaluation and quality assurance of the PMF-results, the retrieval of CSOA seems robust to us.

We agree that V-mode has a limited resolution, and we would not generally use it for determination of N-containing ion fragments. In the particular instance of m/z 84, contributions of C,H,O-containing fragments are rather small, which makes it possible to distinguish the N-containing marker fragment with reasonable certainty even in V-mode. Figure 1 below shows a typical fit on m/z 84, based on a 30 s run. As can be seen, the N-containing fragment can be clearly distinguished. We can add this or a similar figure to the revised manuscript or a supplementary information, if desired.

[Figure]

Figure 1: m/z 84 of a raw spectrum for a 30 s run during a period of increased CSOA concentrations during POPE2014.

Of course the fitting of $C_5H_{10}N^+$ only works as long as CSOA contributes sufficiently to the total m/z 84 signal with respect to contributions from $POA_{noCSOA}$ (primary organic aerosol excluding cigarette smoke), as we discuss in the manuscript (page 25, lines 15-25). We also discussed the limitations of identification and quantification of CSOA using this marker, taking into account the limited resolution of the V-mode (page 25, line 26 to page 26, line 5).

Concerning "the absence of the measurements of molecular markers for cigarette smoking" we identified the fragment ion $C_5H_{10}N^+$ as an indication of nicotine, which is a molecular marker. While we do not have independent measurements of the same marker using

different methods, we used the very good correlation of the time series of this marker ion and the CSOA factor to associate this marker ion with cigarette smoke. The fact that the marker ion was also found in laboratory measurements of cigarette smoke (see discussion on page 24, lines 16-19) gives us confidence in its applicability.

Concerning the referee's comment on the time series: If the time series are affected by boundary layer dynamics (we won't speculate on whether this is the case or not), the "true" diurnal patterns would have an even larger amplitude since boundary layer dynamics would result in inverse structures to the observed patterns. Since the observed diurnal (and weekly) patterns of the CSOA factor agree very well with our observations of smoking activities in the vicinity of the sampling site, we are very confident that the patterns presented in Figure 13 reflect the concentrations of smoking-related aerosol well. For this reason we stated in the text:

> Page 24: "The diurnal cycle of the CSOA factor strongly correlates with typical working hours at the measurement location…" and "show distinct differences between working days and weekend, when the administration of the hospital where the measurements took place was closed, supporting the attribution of this PMF factor to locally emitted CSOA."

Time series of CSOA concentrations show the same as the information summarized in Figure 13, and therefore was omitted in the original manuscript in order to avoid duplicate information. However, we can include the time series in a supplement to our revised submission, as part of a discussion of the factor analysis results.

Comment: (5) the new particle formation in this study appears to have problems too. At least from the average diurnal cycles in Figure 5, we didn't see "banana" shape. On the other hand, the diurnal cycles appears to indicate strong local sources at both sites.

Reply:

Banana-like shapes in particle size distributions are only expected when observing the evolution of the particle size distribution of freshly nucleated particles over time. This can only be done if either the measurement follows the air mass in a Lagrangian experiment or the nucleation occurs over a sufficiently large area simultaneously (like in the boreal forest) and the air masses passing by the measurement site all have the same history. Both are not the case in our measurements, and in similar measurements in urban environments. Therefore it is not surprising that we do not see banana-like shapes in Figure 5 and in the raw data. Also the studies performed by Brines et al. (2015) in the cities of Barcelona, Brisbane, Los Angeles, Madrid, and Rome showed nucleation bursts without strong subsequent growth (depicted in Fig. 5 Brines et al., 2015), different from the typical "banana-like" nucleation episodes usually described in regional background environments. Minguillon et al. (2015) reported the restriction of nucleation events in Barcelona to midday and early afternoon, when condensation sinks are low due to decreased traffic emissions. Therefore, nucleation events found in urban environments are often similar to those found in our study, and do not necessarily exhibit a "banana shape".

Our measurements were performed stationary in a suburban/urban area, which is why we observe a strong contribution of small particles during the rush hour times in the morning

and the evening (of course there are strong local sources at both sites), as discussed in the manuscript (page 15, lines 9-10). During midday, when concentrations of particles from traffic are typically at their minimum, we observe different patterns in the diurnal cycles of particle number concentrations between late fall and late spring. Only during the warmer season we observed increased particle number concentrations during midday. The connection of the increased particle number concentrations during midday with new particle formation events is supported by mean particle number size distributions measured with the FMPS. These indicate increased concentrations of rather small particles during this time of the day, as discussed in the manuscript (pages 15-17).

Comment: (6) the classifications of "home-made" and "advected" might also have large uncertainties. For example, OOA can be from both sources since SV-OOA and LV-OOA cannot be separated. Although nitrate has a shorter life time than sulfate and LV-OOA, many studies have shown that regional transport can be important. I understand the authors can judge this based on the polar plots in Figure 8. In fact, I suggest that the authors re-analyze the polar plots by considering the influences of the number points in each cell. For example, the wind rose plots in Figure 1b shows a small frequency from the northeast, the polar plots in this direction can be significantly biased by sporadic spikes.

Reply:

We agree that this classification has large uncertainties. Therefore we described the comparison of "home-made" and "advected" submicron aerosol as a "rough estimate" in the first sentence of the related paragraph (page 11, line 14). We agree that OOA (not separated into LV- and SV-OOA during Oct/Nov 2013) can be from both, local and remote sources. However, as shown in Figure 7, OOA was found to be rather aged and therefore was assumed to preferentially be associated with the "advected" type. This assumption has some uncertainties and will lead to a small additional uncertainty in the final result of this analysis. However, this small uncertainty is much smaller than the uncertainty implicitly claimed in the text by expressions such as "rough estimate" and "approximately half of the locally measured $PM_1$ was home-made" (page 27, line 26). The same is true for potential small fractions of transported nitrate, which also could contribute a slight additional uncertainty to the overall analysis.

The referee is right that for the northeast wind direction a relatively low number of data points is available due to the low frequency of wind from this direction. However, an estimation shows that this is not a real problem for our interpretation of the data: According to Figure 1b about 5% of the data are associated with wind from this direction. For ca. 15 days of measurement and 1-minute data this corresponds to about 1080 data points (15 x 24 x 60 x 0.05) for this wind direction. These data points distribute over approximately 90 pixels of the polar plots, resulting in an average of ~12 data points per pixel. Furthermore, in the analysis of the polar plots only the general trends were investigated and not individual "hot spots" of single pixels with extraordinarily high numbers (outliers). Therefore we conclude that even for such wind directions a reasonable data base is available to avoid significant biases by sporadic spikes.

**References**

Brines, M., Dall'Osto, M., Beddows, D. C. S., Harrison, R. M., Gomez-Moreno, F., Nunez, L., Artinano, B., Costabile, F., Gobbi, G. P., Salimi, F., Morawska, L., Sioutas, C., and Querol, X.: Traffic and nucleation events as main sources of ultrafine particles in high-insolation developed world cities, Atmos. Chem. Phys., 15, 5929-5945, 10.5194/acp-15-5929-2015, 2015.

Canagaratna, M. R., Jimenez, J. L., Kroll, J. H., Chen, Q., Kessler, S. H., Massoli, P., Ruiz, L. H., Fortner, E., Williams, L. R., Wilson, K. R., Surratt, J. D., Donahue, N. M., Jayne, J. T., and Worsnop, D. R.: Elemental ratio measurements of organic compounds using aerosol mass spectrometry: characterization, improved calibration, and implications, Atmos. Chem. Phys., 15, 253-272, 10.5194/acp-15-253-2015, 2015.

Crippa, M., DeCarlo, P. F., Slowik, J. G., Mohr, C., Heringa, M. F., Chirico, R., Poulain, L., Freutel, F., Sciare, J., Cozic, J., Di Marco, C. F., Elsasser, M., Nicolas, J. B., Marchand, N., Abidi, E., Wiedensohler, A., Drewnick, F., Schneider, J., Borrmann, S., Nemitz, E., Zimmermann, R., Jaffrezo, J. L., Prevot, A. S. H., and Baltensperger, U.: Wintertime aerosol chemical composition and source apportionment of the organic fraction in the metropolitan area of Paris, Atmos. Chem. Phys., 13, 961-981, 10.5194/acp-13-961-2013, 2013a.

Crippa, M., El Haddad, I., Slowik, J. G., DeCarlo, P. F., Mohr, C., Heringa, M. F., Chirico, R., Marchand, N., Sciare, J., Baltensperger, U., and Prevot, A. S. H.: Identification of marine and continental aerosol sources in Paris using high resolution aerosol mass spectrometry, J. Geophys. Res.-Atmos., 118, 1950-1963, 10.1002/jgrd.50151, 2013b.

Faber, P., Drewnick, F., Veres, P. R., Williams, J., and Borrmann, S.: Anthropogenic sources of aerosol particles in a football stadium: Real-time characterization of emissions from cigarette smoking, cooking, hand flares, and color smoke bombs by high-resolution aerosol mass spectrometry, Atmos. Environ., 77, 1043-1051, 10.1016/j.atmosenv.2013.05.072, 2013.

Kostenidou, E., Florou, K., Kaltsonoudis, C., Tsiflikiotou, M., Vratolis, S., Eleftheriadis, K., and Pandis, S. N.: Sources and chemical characterization of organic aerosol during the summer in the eastern Mediterranean, Atmos. Chem. Phys., 15, 11355-11371, 10.5194/acp-15-11355-2015, 2015.

Minguillon, M. C., Brines, M., Perez, N., Reche, C., Pandolfi, M., Fonseca, A. S., Amato, F., Alastuey, A., Lyasota, A., Codina, B., Lee, H. K., Eun, H. R., Ahn, K. H., and Querol, X.: New particle formation at ground level and in the vertical column over the Barcelona area, Atmospheric Research, 164, 118-130, 10.1016/j.atmosres.2015.05.003, 2015.

Ng, N. L., Canagaratna, M. R., Zhang, Q., Jimenez, J. L., Tian, J., Ulbrich, I. M., Kroll, J. H., Docherty, K. S., Chhabra, P. S., Bahreini, R., Murphy, S. M., Seinfeld, J. H., Hildebrandt, L., Donahue, N. M., DeCarlo, P. F., Lanz, V. A., Prevot, A. S. H., Dinar, E., Rudich, Y., and Worsnop, D. R.: Organic aerosol components observed in Northern Hemispheric datasets from Aerosol Mass Spectrometry, Atmos. Chem. Phys., 10, 4625-4641, 10.5194/acp-10-4625-2010, 2010.

Ortega, A. M., Hayes, P. L., Peng, Z., Palm, B. B., Hu, W. W., Day, D. A., Li, R., Cubison, M. J., Brune, W. H., Graus, M., Warneke, C., Gilman, J. B., Kuster, W. C., de Gouw, J., Gutierrez-Montes, C., and Jimenez, J. L.: Real-time measurements of secondary organic aerosol formation and aging from ambient air in an oxidation flow reactor in the Los Angeles area, Atmos. Chem. Phys., 16, 7411-7433, 10.5194/acp-16-7411-2016, 2016.

Sun, Y., Du, W., Fu, P., Wang, Q., Li, J., Ge, X., Zhang, Q., Zhu, C., Ren, L., Xu, W., Zhao, J., Han, T., Worsnop, D. R., and Wang, Z.: Primary and secondary aerosols in Beijing in winter: sources, variations and processes, Atmos. Chem. Phys., 16, 8309-8329, 10.5194/acp-16-8309-2016, 2016.

Xu, L., Williams, L. R., Young, D. E., Allan, J. D., Coe, H., Massoli, P., Fortner, E., Chhabra, P., Herndon, S., Brooks, W. A., Jayne, J. T., Worsnop, D. R., Aiken, A. C., Liu, S., Gorkowski, K., Dubey, M. K., Fleming, Z. L., Visser, S., Prevot, A. S. H., and Ng, N. L.: Wintertime aerosol chemical composition, volatility, and spatial variability in the greater London area, Atmos. Chem. Phys., 16, 1139-1160, 10.5194/acp-16-1139-2016, 2016.

Zhang, J. K., Sun, Y., Liu, Z. R., Ji, D. S., Hu, B., Liu, Q., and Wang, Y. S.: Characterization of submicron aerosols during a month of serious pollution in Beijing, 2013, Atmos. Chem. Phys., 14, 2887-2903, 10.5194/acp-14-2887-2014, 2014.

Zhang, Q., Jimenez, J. L., Canagaratna, M. R., Ulbrich, I. M., Ng, N. L., Worsnop, D. R., and Sun, Y.: Understanding atmospheric organic aerosols via factor analysis of aerosol mass spectrometry: a review, Anal. Bioanal. Chem., 401, 3045-3067, 10.1007/s00216-011-5355-y, 2011.

Zhang, Y. J., Tang, L. L., Wang, Z., Yu, H. X., Sun, Y. L., Liu, D., Qin, W., Canonaco, F., Prevot, A. S. H., Zhang, H. L., and Zhou, H. C.: Insights into characteristics, sources, and evolution of submicron aerosols during harvest seasons in the Yangtze River delta region, China, Atmos. Chem. Phys., 15, 1331-1349, 10.5194/acp-15-1331-2015, 2015.

---

## Referee Comment (RC2) · Anonymous Referee #2 · 24 Sep 2016

The following paper presents a comprehensive study at two different sites of Rome and surroundings (called urban and suburban) for two different seasons, totalizing four rather short datasets. Measurements of coarse and fine aerosol modes were performed, with focus on the latter, as well as several gases and meteorological parameters. Organic aerosol data were further analyzed with PMF and several different components were identified. Overall I recommend the paper if the comments below would be addressed.

Some specific comments, trying not to overlap with the previous Referee:

1) P7, L21-23: Why Canagaratna et al. (2015) was not used instead? 2) P7, L29-30: What would such variabilities be? Are they inside a 20% variation? 3) P8, L3-12: More

information about the PMF analysis would be very welcome. For example, what was the input matrix size, how many rows and columns? Were the outliers downweighted? Same seasons were run together or separately? 4) P8, L13-22: The Q/Qexp value is normally useful in the identification of the final solution. However, no information on that was provided. 5) P10, L15-17: It is true that absolute values of POA mass concentration are larger in the suburban area compared to the urban; however, note that the percentages are smaller. According to Figure 2 POA in the urban is still 58% of organics, while suburban is 46%. So it seems to me that POA mass is larger in the suburban because PM1 is ∼2 times larger and not because BBOA itself. I think those lines need clarification. 6) Table 3 e Figure 2: Beautiful and instructive figure! Please verify the PM1 values, they do not agree with the Table 3. For instance, Figure 2 says that PM1 at suburban/2013 was 12 $\mu$g m-3, while the Table 3 says it was 15±10 $\mu$g m-3. Which one is correct? Same thing to the other 3 campaigns. 7) P10, L22-25: How can a fresher OOA (so called SV-OOA) be extracted if solar radiation and O3 are larger? Shouldn't it be the opposite? The larger amount of solar radiation and air T should fasten the aerosol ageing. Please provide further explanation. For example, could the fresher OOA be maybe related to different air masses reaching the region during this time of the year, further air circulation/dispersion? Also this time of the year may allow for larger ranges in air T and solar radiation, . . . 8) P10, L27-30: The reduced solar radiation makes a reasonable explanation at the urban site, where OOA increased from 42% to 76% from winter to summer. However, not at the suburban where almost no change was observed (53-56%). That is, if the radiation increased at both sites, why only at the urban such effect was observed? For sure solar radiation is important, but it is not the only aerosol ageing agent. There are locations where LV-OOA is dominant under almost no solar radiation (i.e. Finland). Moreover, was fog observed/measured during the Oct/Nov measurements? Particle water uptake associated with fog may promote aerosol processing through heterogeneous-phase chemistry. The presence of fog has been well studied in other regions of Italy during the same season (i.e. Po Valley). 9) P11, L4-6: The authors say that the lower fraction of sulphate on Oct/Nov could be

related to the enhance of POA because of lower boundary layer heights (BLH). If the BLH was lower, that should affect all pollutants equally, right? In addition, a significant change in POA fraction, between seasons, was only observed at the urban site, not in the suburban. 10) P11, L22-25: The authors say that absolute concentrations of suburban home-made PM1 are larger than urban, probably due to meteorological conditions. Right. However, note that in percentage the urban is larger than the suburban (70 and 47%). In reality, home-made PM1 is smaller in absolute concentrations at the suburban because all PM1 is ∼2 times smaller. Thus, the meteorological conditions are the main reason for such decrease in PM1. The home-made PM1 decrease is just following the PM1 trend. It seems to me that the use of absolute concentrations only, can slightly misguide the reader. 11) P13, L6-8: It would be very kind of the authors to point out in here which minerals were identified in Silvia Nara, personal communication 2016, since this communication is not easily available. 12) P18, L16-20: In fact, what Jimenez et al. (2009) state is that the larger dynamic range in solar radiation and air temperature may cause the different OOAs. 13) P19, L23-28: Those last sentences are slightly confusing and seem to contradict the "home-made vs advected" section of this study. That is, in the "home-made vs advected" section nitrate was considered home-made. Here it is said that nitrate is north-east advected. Please clarify. Why are those sections separated if they complement one another? 14) P21, L7-11: I'm surprised BBOA is still so significant in late spring. Unfortunately, the authors did not provide the MS for the reader. Are the two BBOAs (summer-like and winter-like) similar? If one BBOA was more related to agricultural and the other to agricultural and domestic heating, I'd expect to see some differences among them, was any observed? Also with respect to external tracers. 15) P24, L1-4: Very interesting! The MS of CSOA identified by this study look quite different from the previous (Fröhlich et al., 2015 and Faber et al., 2013), with prominent fragments at m/zs 84. Also the N-containing fragment at m/z 42 look prominent and present in NIST MS database. Why this fragment was not mentioned in this study? Was it because of interference from other fragments in the same nominal m/z? 16) P24, L5-6: Just a comment: when comparing SV-OOA

and LV-OOA MS from the AMS database (Ng et al., 2010), also a R2=0.65 is obtained, and those are very different things! Thus, I'd be very careful in comparing CSOA obtained in this study to previous ones. I mean, there is no need to try to make them look similar, since the current study seems to have identified a much clearer CSOA factor. 17) P25, L3: The authors used C5H10N+ fragment as a marker for CSOA. Since the measurements were performed with and HR-ToF-AMS, which provides size distribution data. Did the authors try to identify the size distribution of the CSOA using this same marker? Or perhaps just UMR m/z 84, 42?

Minor comments: 18) P18, L4: The right citation is Alfarra et al., 2004 (Characterization of urban and rural organic particulate in the Lower Fraser Valley using two Aerodyne Aerosol Mass Spectrometers). 19) P18, L5-11: This sentence is too long and difficult to read. 20) P20, L14: Probably the ":" should be replaced by a ".". 21) P23, L13: Replace ":" by ".".
* * *

---

## Referee Comment (RC3) · Anonymous Referee #3 · 18 Oct 2016

This paper describes measurements from campaigns in and around Rome during different seasons. It predominantly considers HR-AMS data but also incorporates estimates of Saharan dust convection and aerosol size distribution. It offers valuable insights relating to the impact of different sources on PM1 in urban areas in Europe and builds on my previous studies. It makes particular advances in the consideration of cigarette smoke and its measurement using HR-AMS. As with many short seasonal campaigns it is difficult to draw firm conclusions due to the large variation in meteorological conditions, however, the authors have contrasted different seasons without over-interpreting. Specific Comments Labelling NO3 as 'home made' (pg 11 line 16). Although the formation of NO3 from gaseous precursors occurs over a short time scale

and will be partially 'home made', it is affected over large distances and a portion is likely to be advected. This will impact on your % estimates and should be discussed and altered appropriately. Two modes in particle size distribution (pg 14 line 1). The second 'peak' in size distribution at 0.6 um may be an influence of reduced counting efficiency to aerosols in this size range. Bivariate polar plots (Fig 8) NH4 is reported for 2013 but not 2014. Technical Comments pg 11 line 4 'lowered' should read 'lower' pg 12 line 3 'always was' should read 'was always' Table 5 'PM10 in PM1' should read 'PM1 in PM10' Pg 17 line 19-20 'probably less precursors are' should read 'lower concentrations of precursors are probably' and 'more precursors' should read higher concentrations of precursors Pg 19 line 26 'aging level' should read 'degree of aging'

---

## Author Comment (AC2) · 14 Nov 2016

Response to Reviewer Comments

We are thankful to the reviewers for their constructive and thoughtful comments that helped improving the manuscript. We have revised the manuscript according to these comments and added a supplement including further information on the AMS and PMF data quality assurance. Listed below is our point-by-point response to each of the reviewers' comments (red).

In addition, we received revised information on the time interval defined as "dust" period from the EC-LIFE+ DIAPASON project team: As a consequence of results from additional PIXE analysis they have changed the "dust" period for the DIAPASON2013 field campaign to 23.10.-01.11.2013 (before: 29.10.-01.11.2013) (Barnaba et al., in preparation 2016). This period includes days (23.-29.10.13) with rather low concentrations of coarse particles; however PIXE results and model calculations clearly indicate the presence of dust at ground level during these days. To avoid discrepancies and to allow for a better comparability between our results and those from their measurements, we therefore have adapted the period of the 2013 dust event in our manuscript and have revised our analysis in Sect. 4.2.1 accordingly. The no dust period is now defined as the period 02.11.-07.11.13 (before: 23.-28.10 + 02.11.-07.11.13). Extending the period of the dust event in 2013 did not cause any substantial changes in the conclusion. With respect to the background conditions, the $PM_{10}$ fraction was increased (from 13 to 17 µg m$^{-3}$), whereas the coarse particle fraction ($PM_{10-2.5}$) was decreased (from 3.1 to 2.4 µg m$^{-3}$). While absolute concentrations of the AMS species have changed for the dust (increase of 5%) and no dust (decrease of 50%) period, the relative composition remains nearly the same. The strong decrease of the $PM_1$ fraction observed for the "new" no dust period likely results from a change in air mass origin. In the "old" definition the no dust period was composed of two time intervals, one before and one after the dust event, whereas in the "new" definition the no dust period only includes the period after the dust event. In this "new" no dust period, a change of air masses (Mediterranean region to Atlantic Ocean) occurred. While the "old" no dust period was influenced by air masses both from the Mediterranean region and the Atlantic Ocean, the "new" no dust period is preferentially influenced by air masses from the Atlantic Ocean. Furthermore, during the no dust period higher wind speeds were observed compared to the dust period, resulting in stronger dilution effects. This is briefly discussed in the revised manuscript (P15, L19-22).

All changes that have been made to the manuscript can be tracked in the attached version. When referring to the manuscript, we always refer to the page and line number of the revised manuscript including the track changes.

**Response to anonymous referee #1**

We thank anonymous referee #1 for his/her helpful comments.

**General comments:**

Comment: The manuscript by Struckmeier et al. analyzed four datasets that were collected at two sites in different seasons in Rome (suburban vs. urban). This study contains various real-time online measurements including aerosol chemical composition, gaseous species,

particle number concentrations, and meteorological parameters. The sources of organic aerosols (OA) were also analyzed with positive matrix factorization. While rich chemical information was provided in this work to address the sources, dynamics, and spatial variations, the discussions e.g., composition, dust event, new particle formation, OA, and CSOA are scattered and lack of focus.

Reply: The intention and focus of our manuscript is to provide an assessment of the sub-micron aerosol and its potential sources at an urban and a suburban location in the Rome area during different measurement periods. This includes a general overview (the "rich chemical information" mentioned by the reviewer), but also deeper discussion on specific areas, which need to be addressed in order to characterize a complex urban environment like Rome. In particular with respect to the very different sources it would have been incomplete if for example the Saharan dust or the NPF, or other aspects had been left out. Thus, the focus of our work is on the description and, if possible, explanation of aerosols from the various sources, similar to approaches in previous publications from different groups (e.g., Crippa et al., 2013b; Kostenidou et al., 2015) which show that there is a general interest in this kind of analysis.

**Specific comments:**

Comment: Also, I have several major concerns on data analysis and the interpretations: (1) each campaign lasted less than two weeks, and most importantly, the measurements at the suburban and urban sites were not simultaneous. This clearly increases the uncertainties in comparing aerosol chemistry and sources between the two sites. In addition, it is difficult to see the dynamic variations of aerosol species in Rome if the authors didn't present time series data.

Reply: The durations of the data acquisition periods are in detail:

DIAPASON2013: 16 days of measurements (including 14 full days)

POPE2013: 8 days of measurements (including 6 full days)

DIAPASON2014: 16 days of measurements (including 14 full days)

POPE2014: 14 days of measurements (including 12 full days)

This shows that only POPE2013 was shorter than two weeks. The measurements were performed subsequently and not simultaneously as stressed in the manuscript, e.g., in the first sentence of the abstract. Obviously this results in a larger uncertainty when comparing the observations from the urban and suburban site, as the reviewer correctly states. This is why we do not generalise observations or differences in observations at different locations or during different seasons, but rather present our findings considering potentially different weather conditions etc., as clearly indicated several times in the manuscript, e.g.:

- P12, L33-P13, L3: "Regarding absolute $PM_1$ concentrations […] neither any general conclusion whether aerosol mass concentrations are higher at the city centre or in the suburb, nor whether $PM_1$ concentrations are elevated during any of the two different seasons, can be drawn."
- P13, L5-7: "As discussed above, changes in meteorological conditions are likely one explanation for this result …"

- L13, L12-14: "In addition to meteorological conditions (e.g. solar radiation, BLH, TKE, air mass origin, etc.) local air quality can be strongly influenced by local emission from various sources (traffic, cooking, biomass burning)."

Nevertheless, each of the four measurement intervals provides valuable information on properties and dynamics of various aerosol types in the city centre and at the suburban location, which are found to be sufficiently robust within the available measurement time.

In our manuscript we focus on general features of the observed aerosols. Therefore we did not show time series of aerosol variables. In order to allow the reader to get a picture on temporal variations during the field measurements, we added a supplement presenting time series of all PMF factors and some selected important other aerosol variables.

Comment: (2) the data quality was not validated adequately, particularly the AMS measurements. A simple comparison between PM1 measured by EDM and that measured by AMS and MAAP (NR-PM1 + BC) will help.

Reply: We agree that assuring data quality is good practice. During the analysis of the AMS (and other instruments') data all standard procedures, checks, calibrations, corrections, intercomparisons, etc. have been performed (see statements in Sections 3.1 and 3.2). Besides other necessary quality checks, also comparison of AMS plus BC data with independent $PM_1$ data (EDM, but also $PM_1$ calculated from size distribution measurements) was performed as well as many other comparisons. All comparisons showed reasonable agreement (e.g., for $PM_1$ from AMS+BC versus EDM, $R^2$=0.69-0.85, Slopes: 0.73-1.37). We included the time series and scatter plots of AMS+BC and EDM $PM_1$ in the supplement, as well as a short discussion of the intercomparisons.

Comment: (3) the AMS data analysis needed to be expanded. For example, which approach (Aiken et al., 2008 or Canagaratna et al., 2015) was used to calculate the elemental ratios the calculation of elemental ratios? If there are elemental ratios, why did the authors still use f43 and f44 to discuss the oxidation states?

Reply: The results from AMS measurements were inspected for relationships among these data and relationships with data from other instruments. For the data obtained with the AMS and the other instruments temporal evolution, diurnal patterns and individual events were analysed (see Section 4).

Elemental ratios were calculated based on the current state of the art method (Canagaratna et al., 2015). We thank the reviewer for pointing out this missing information, which we added to the revised manuscript (P7, L22-23).

Our discussion using f43 and f44 focused on the aerosol aging levels, for which the "triangle" introduced by Ng et al. (2010) is commonly used (e.g., Ortega et al., 2016; Xu et al., 2016; Zhang et al., 2015). We also calculated elemental ratios for the various factors, but didn't include them in the discussion on aerosol aging levels since they show equivalent behaviour as the f44/f43 data. For more completeness, we added the elemental ratios of each PMF factor to the supplement.

Comment: (4) the PMF analysis is a big weakness of this study. The authors didn't have a full evaluation of the PMF results. At least, the authors need to present the mass spectral profiles and times series of all OA factors, and also the comparisons with collocated measurements.

Reply: The performed PMF analysis and corresponding tests are subject of Section 3.2. All typical and many other tests to validate the results were applied, amongst others those according to the guidelines by Zhang et al. (2011), Table 1. All relevant information (e.g. m/z-range used for PMF, method of error and data matrix preparation, treatment of isotopes, treatment of low S/N data, treatment of $CO_2$-related ions, range of factors, fPeak and seed investigated) is detailed in the text.

Furthermore, each PMF factor was inspected for reasonability and validated by comparing factor time series with external species and mass spectral profiles with such from literature. Details are given in the text, e.g., P8, L18-20; P20, L25-29; P21, L17-23; P23, L5-9; P24, L5-8; P25, L1 -P26, L4.

So far we did not present all individual correlation plots and mass spectra in graphs, but presented this information in the form of $R^2$ values to save journal space and keep the length of the manuscript at a reasonable level without losing information. For more completeness we now added the mass spectral profiles and the time series of all PMF factors, as well as the time series of collocated measurements of tracer species and their correlations with PMF factors to the supplement.

Comment: The diurnal correlations the authors mentioned in page 21 did not mean much.

Reply: Such correlations are commonly used to show potential relationships between variables (e.g., Sun et al., 2016; Zhang et al., 2014), and to validate PMF factors (Zhang et al., 2011). In P23 we discuss that the diurnal patterns of HOA correlate well with those of other typically traffic-related species (BC, NOx, PAH). As stated in P23, L8-9, not only the diurnal cycles, but also the time series correlate well. To allow the reader to check this also for the temporal variations in the original time series, the time series of HOA and BC were also added to the supplement.

Comment: Figure 2 also showed substantial differences in HOA/BC ratios at the two sites in different seasons, and surprising BC contributions, which should be well interpreted.

Reply: We thank the reviewer for this valuable comment, and added some more discussion on this topic. We agree that there are differences in the HOA/BC ratios observed during the fall 2013 campaigns and during the spring 2014 campaigns (fall 2013: 0.26-0.33; spring 2014: 0.46-0.53). We attribute this to the fact that both, biomass burning and traffic contribute to total BC, as mentioned on P10, L29-30. Biomass burning is more predominant in fall, leading to higher BC mass concentrations, and therefore lower HOA/BC ratios during this season. The fact that larger ratios were found in the city centre in both seasons reflects the lower contribution of biomass burning emissions at this site. The contribution of different sources (traffic and biomass burning) also explains why the ratio of HOA/BC from our study is different to such ratios found in source measurements of pure traffic-related emissions.

Separation of BC related to the two sources unfortunately is not possible with the instrument used (MAAP), but measurements in other cities with the aethalometer instrument (e.g., Crippa et al., 2013a) have shown that BC contributions from biomass burning to total BC can be non-negligible. In the revised manuscript we extended the discussion of the aerosol composition and added to Sect. 4.1 (P10, L31-P11, L4):

"Ratios of HOA/BC (DIAPASON2013: 0.26, POPE2013: 0.33, DIAPASON2014: 0.46, POPE2014: 0.53) were lower during fall 2013, indicating the more dominant contribution of biomass burning emissions to BC during this time. The fact that contributions from biomass burning to total BC concentrations are non-negligible was also found earlier (e.g. Crippa et al. (2013a). Generally, the HOA/BC ratios were higher in the urban compared to the suburban location in each year. This reflects the stronger influence of traffic emissions and the lower contribution of biomass burning emissions in the city centre."

Comment: The PMF uncertainties lead to another major concern of the cigarette smoking factor. Although the authors concluded this as a major finding and presented a long discussion on it, it is still not convincing due to the limited resolution of V-mode (C5H10N+) and the absence of the measurements of molecular markers for cigarette smoking. I am also suspicious that the diurnal profile of CSOA did not reflect cigarette smoking that is expected less affected by boundary layer dynamics (if the authors claimed it as a point source). Showing the times series of CSOA factor will help.

Reply: PMF analysis is always associated with uncertainties. As described in the manuscript we have taken all care in order to minimize these uncertainties and we have made multiple tests and comparisons with other data (e.g. mass spectra of previously identified CSOA, as described in the manuscript and now also shown in the added supplementary information) in order to obtain results as robust as possible. While the CSOA factor was first obtained from the PMF analysis of the whole mass spectra, afterwards the $C_5H_{10}N^+$ marker was identified and found to have time series that correlate very well with those of CSOA. Indeed, as discussed in the manuscript, CSOA has been identified before from AMS data in which the newly found marker fragment at m/z 84 was not fitted at all (Faber et al., 2013) and yielded a very similar factor mass spectrum with correlation coefficients $0.65 < R^2 < 0.96$ (P25, L31). (Re-analysis later showed that the marker fragment indeed was present also in these data, as discussed in the manuscript, P26 L9-11.) Therefore, and from other tests we performed in the data evaluation and quality assurance of the PMF-results, the retrieval of CSOA seems robust to us.

We agree that V-mode has a limited resolution, and we would not generally use it for determination of N-containing ion fragments. In the particular instance of m/z 84, contributions of C,H,O-containing fragments are rather small, which makes it possible to distinguish the N-containing marker fragment with reasonable certainty even in V-mode. Figure S16 in the supplementary information shows a typical fit on m/z 84, based on a 30 s run. As can be seen, the N-containing fragment can be clearly distinguished.

Of course the separation and fitting of $C_5H_{10}N^+$ only works as long as CSOA contributes sufficiently to the total m/z 84 signal with respect to contributions from POA$_{noCSOA}$ (primary organic aerosol excluding cigarette smoke), as we discuss in the manuscript (P27,L7-17). We

also discussed the limitations of identification and quantification of CSOA using this marker, taking into account the limited resolution of the V-mode (P27, L26-P28, L6).

Concerning "the absence of the measurements of molecular markers for cigarette smoking" we identified the fragment ion $C_5H_{10}N^+$ as an indication of nicotine, which is a molecular marker. While we do not have independent measurements of the same marker using different methods, we used the very good correlation of the time series of this marker ion and the CSOA factor to associate this marker ion with cigarette smoke. The fact that the marker ion was also found in laboratory measurements of cigarette smoke (see discussion on P26, L6-11) gives us confidence in its applicability.

Concerning the referee's comment on the time series / diurnal profile: If the time series are affected by boundary layer dynamics (we won't speculate on whether this is the case or not), the "true" diurnal patterns would have an even larger amplitude since boundary layer dynamics would result in inverse structures to the observed patterns. Since the observed diurnal (and weekly) patterns of the CSOA factor agree very well with our observations of smoking activities in the vicinity of the sampling site, we are very confident that the patterns presented in Figure 13 reflect the concentrations of smoking-related aerosol well. For this reason we stated in the text:

> P26, L13-19: "The diurnal cycle of the CSOA factor strongly correlates with typical working hours at the measurement location…" and "show distinct differences between working days and weekend, when the administration of the hospital where the measurements took place was closed, supporting the attribution of this PMF factor to locally emitted CSOA."

For completeness we also provide time series of CSOA and the nicotine tracer ion concentrations in the supplement (Figures S9 and S14).

Comment: (5) the new particle formation in this study appears to have problems too. At least from the average diurnal cycles in Figure 5, we didn't see "banana" shape. On the other hand, the diurnal cycles appears to indicate strong local sources at both sites.

Reply: Banana-like shapes in time series of particle size distributions are only expected when observing the evolution of the particle size distribution of freshly nucleated particles over time. This can only be done if either the measurement follows the air mass in a Lagrangian experiment or the nucleation occurs over a sufficiently large area simultaneously (like in the boreal forest) and the air masses passing by the measurement site all have the same history. Both are not the case in our measurements, and in similar measurements in urban environments. Therefore it is not surprising that we do not see banana-like shapes in Figure 5 and in the raw data. Also the studies performed by Brines et al. (2015) in the cities of Barcelona, Brisbane, Los Angeles, Madrid, and Rome showed nucleation bursts without strong subsequent growth (depicted in Fig. 5 Brines et al., 2015), different from the typical "banana-like" nucleation episodes usually described in regional background environments. Minguillon et al. (2015) reported the restriction of nucleation events in Barcelona to midday and early afternoon, when condensation sinks are low due to decreased traffic emissions. Therefore, nucleation events found in urban environments are often similar to those found in our study, and do not necessarily exhibit a "banana shape".

Our measurements were performed stationary in a suburban/urban area, which is why we observe a strong contribution of small particles during the rush hour times in the morning and the evening (of course there are strong local sources near both sites), as discussed in the manuscript (P16, L12-13). During midday, when concentrations of particles from traffic are typically at their minimum, we observe different patterns in the diurnal cycles of particle number concentrations between late fall and late spring. Only during the warmer season we observed increased particle number concentrations during midday. The connection of the increased particle number concentrations during midday with new particle formation events is supported by average particle number size distributions measured with the FMPS. These indicate increased concentrations of rather small particles during this time of the day, as discussed in the manuscript (P16-18).

Comment: (6) the classifications of "home-made" and "advected" might also have large uncertainties. For example, OOA can be from both sources since SV-OOA and LV-OOA cannot be separated. Although nitrate has a shorter life time than sulfate and LV-OOA, many studies have shown that regional transport can be important. I understand the authors can judge this based on the polar plots in Figure 8. In fact, I suggest that the authors re-analyze the polar plots by considering the influences of the number points in each cell. For example, the wind rose plots in Figure 1b shows a small frequency from the northeast, the polar plots in this direction can be significantly biased by sporadic spikes.

Reply: We agree that this classification has large uncertainties. Therefore we described the comparison of "home-made" and "advected" submicron aerosol as a "rough estimate" in the first sentence of the related paragraph (P12, L1). We agree that OOA (not separated into LV- and SV-OOA during Oct/Nov 2013) can be from both, local and remote sources. However, as shown in Figure 7, OOA was found to be rather aged and therefore was assumed to preferentially be associated with the "advected" type. This assumption has some uncertainties and will lead to a small additional uncertainty in the final result of this analysis. However, this small uncertainty is much smaller than the uncertainty implicitly claimed in the text by expressions such as "rough estimate" and "approximately half of the locally measured $PM_1$ was home-made" (P29, L28). The same is true for potential small fractions of transported nitrate, which also could contribute a slight additional uncertainty to the overall analysis, thereby however partially balancing the potential OOA-related bias.

The referee is right that for the northeast wind direction a relatively low number of data points is available due to the low frequency of wind from this direction. However, an estimation shows that this is not a real problem for our interpretation of the data: According to Figure 1b about 5% of the data are associated with wind from this direction. For ca. 15 days of measurement and 1-minute data this corresponds to about 1080 data points (15 x 24 x 60 x 0.05) for this wind direction. These data points distribute over approximately 90 pixels of the polar plots, resulting in an average of ~12 data points per pixel. Furthermore, in the analysis of the polar plots only the general trends were investigated and not individual "hot spots" of single pixels with extraordinarily high numbers (outliers). Therefore we conclude that even for such wind directions a reasonable data base is available to avoid significant biases by sporadic spikes.

Comment: With this, I cannot recommend it for publication on ACP.

Reply: We hope that after answering to the referee's comments as well as including the requested further information to the manuscript (e.g. discussion of HOA/BC ratios) and adding a supplement with detailed information on the AMS (e.g. comparison of total AMS+BC vs. $PM_1$ (EDM)) and PMF (e.g. factor time series, factor spectra, comparisons of PMF factors with external tracers) quality assurance, we can overcome the referee's concerns.

**Response to anonymous referee #2**

We thank anonymous referee #2 for his/her thoughtful and helpful comments. We especially appreciate the suggestions concerning the section on cigarette emissions.

**General comments:**

The following paper presents a comprehensive study at two different sites of Rome and surroundings (called urban and suburban) for two different seasons, totalizing four rather short datasets. Measurements of coarse and fine aerosol modes were performed, with focus on the latter, as well as several gases and meteorological parameters. Organic aerosol data were further analyzed with PMF and several different components were identified. Overall I recommend the paper if the comments below would be addressed.
Some specific comments, trying not to overlap with the previous Referee:

**Specific comments:**

  1)  P7, L21-23: Why Canagaratna et al. (2015) was not used instead?

Reply: Many thanks to the referee for providing that helpful comment. Even if not mentioned, elemental analysis indeed was performed based on the improved-ambient method introduced by Canagaratna et al. (2015). This information was added in the revised manuscript (P7, L22-23).

  2)  P7, L29-30: What would such variabilities be? Are they inside a 20% variation?

Reply: The variabilities are inside a 20% variation. IE calibrations performed before and after the campaigns in 2013 varied by ~10% (IE: $1.9*10^{-7}$, $1.95*10^{-7,}$ $2.28*10^{-7}$) and we found a variation of ~20% around the average for the IE calibrations performed in 2014 (IE: $1.89*10^{-7}$, $2.64*10^{-7}$). The information on the variability of the IE values for each year was added to the revised manuscript (P7, L30-32):

"Therefore, the observed variability of the IE values (2013: ~10%, 2014: ~20%) is assumed to stem only from the uncertainty of the calibrations, and for each year averages of the determined IE and RIE values were used for data analysis."

3+4) P8, L3-12: More information about the PMF analysis would be very welcome. For example, what was the input matrix size, how many rows and columns? Were the outliers downweighted? Same seasons were run together or separately?

and:

P8, L13-22: The Q/Qexp value is normally useful in the identification of the final solution. However, no information on that was provided.

Reply: In the PMF analysis, each measurement campaign (DIAPASON2013, POPE2013, DIAPASON2014, POPE2014) was evaluated separately. PMF input data were treated as proposed by Ulbrich et al. (2009), as stated at P8, L3-13. M/z with signal-to-noise ratios (SNR) < 0.2 were removed, m/z with 0.2<SNR>2 were downweighted by increasing their error by a factor of 2 and m/z 44-related peaks were downweighted by a factor of sqrt(5).

More information on the PMF analysis (matrix size of the input data, the $Q/Q_{exp}$ values, number of factors for the chosen solution, factor mass spectra, factor and tracer time series) is now provided in the supplementary information.

5) P10, L15-17: It is true that absolute values of POA mass concentration are larger in the suburban area compared to the urban; however, note that the percentages are smaller. According to Figure 2 POA in the urban is still 58% of organics, while suburban is 46%. So it seems to me that POA mass is larger in the suburban because PM1 is 2 times larger and not because BBOA itself. I think those lines need clarification.

Reply: The reviewer is completely right, during 2013 the absolute and the relative contributions of POA-related aerosol at the two sites show an opposite behaviour. This is mainly due to the massive differences in OOA mass concentration measured at the two sites. We assume this OOA is mostly advected, i.e. the observed changes in OOA concentrations between the two sites are due to changes in the histories of the air masses sampled during the different measurement periods, which is also apparent e.g. from the different $SO_4$ mass concentrations. This change in air mass history, however, should not affect the primary emitted species. Therefore, comparisons of relative POA contributions would be biased by changes in sampled air masses (where OOA absolute mass concentration would change, while POA would not). For the comparison of these locally produced aerosol species, we therefore decided to rather focus on absolute POA concentrations, in order to avoid such an influence of sampled air mass. In order to make the reasoning behind this selection clearer and more transparent we added a brief discussion about this to the revised manuscript (P10, L24-29):

"Note that due to the fact that OOA concentrations are significantly higher in the suburban site in 2013, the relative contribution of POA-related aerosol types (i.e. HOA, COA, and BBOA) is lower at the suburban location during this year. However, in order to avoid a bias in the comparison of the more locally generated POA-related aerosol types by advected aerosol mass, we compare the absolute mass concentrations for the different aerosol types at the different measurement locations/times."

6) Table 3 e Figure 2: Beautiful and instructive figure! Please verify the PM1 values, they do not agree with the Table 3. For instance, Figure 2 says that PM1 at suburban/2013 was 12 µg m-3, while the Table 3 says it was 15±10 µg m-3. Which one is correct? Same thing to the other 3 campaigns.

Reply: The discrepancy between the $PM_1$ concentrations reported in Table 3 and Figure 2 of the manuscript results from the 2 different measurement instruments these concentrations are based on. $PM_1$ concentrations presented in the table originate from EDM measurements (optical particle detection), whereas the $PM_1$ concentrations presented in Figure 2 represent the sum of BC (from MAAP-measurements) and total AMS concentration. The difference in the $PM_1$ concentrations can arise from the slightly different measurement size ranges or measured particle components, e.g. the AMS+BC $PM_1$ does not include non-refractory species like Saharan dust, the AMS has a lower cut-off than the EDM, etc. (see discussion in the supplementary material). In the revised manuscript we added a footnote to Table 3, stating that the PM1/2.5/10 values are from EDM measurement. Furthermore, we added also the AMS+BC PM1 concentrations to Table 3 in the revised manuscript and present a comparison of AMS+BC and $PM_1$ from EDM measurements together with a discussion of the differences in the supplement.

7) P10, L22-25: How can a fresher OOA (so called SV-OOA) be extracted if solar radiation and O3 are larger? Shouldn't it be the opposite? The larger amount of solar radiation and air T should fasten the aerosol ageing. Please provide further explanation. For example, could the fresher OOA be maybe related to different air masses reaching the region during this time of the year, further air circulation/dispersion? Also this time of the year may allow for larger ranges in air T and solar radiation, …

Reply: The reviewer is absolutely right, increased solar radiation and $O_3$ should result in faster aerosol (and precursor) ageing. This exactly is the reason why during the 2014 measurements precursor material emitted in the greater Rome area are likely sufficiently quickly oxidized to form the fresh SV-OOA before arrival at our measurement site. During the 2013 measurements oxidation processes are slower such that apparently no (or not enough) freshly produced secondary aerosol from precursors from the greater Rome area can be found at the measurement sites but only much more aged material that is transported from further away.

To make this clearer we added the following statement to the revised manuscript (P11, L8-9): "This SV-OOA likely is the result of quick formation of secondary aerosol from precursors originating from the greater Rome area."

8) P10, L27-30: The reduced solar radiation makes a reasonable explanation at the urban site, where OOA increased from 42% to 76% from winter to summer. However, not at the suburban where almost no change was observed (53-56%). That is, if the radiation increased at both sites, why only at the urban such effect was observed? For sure solar radiation is important, but it is not the only aerosol ageing agent. There are locations where LV-OOA is dominant under almost no solar radiation (i.e. Finland). Moreover, was fog observed/measured during the Oct/Nov measurements? Particle water uptake associated

with fog may promote aerosol processing through heterogeneous-phase chemistry. The presence of fog has been well studied in other regions of Italy during the same season (i.e. Po Valley).

Reply: In the lines cited by the reviewer, we stated that the reason for the observation of only a single OOA factor (a rather aged OOA, which is similar to the LV-OOA observed during the spring measurements) in fall is due to the slower processing of precursor material (due to less solar radiation) and therefore reduced production of very fresh oxidized material (see also reply to comment #7). We do not speculate about the reasons for differences or missing differences in absolute concentrations of total oxidized material or their fractional contribution to total (organic) aerosol.

We completely agree that there are other potential factors than solar radiation that could influence the generation of oxidized material. During our measurements we did not observe distinct fog events or other influences which could potentially dominate the formation and aging of OOA.

9) P11, L4-6: The authors say that the lower fraction of sulphate on Oct/Nov could be related to the enhance of POA because of lower boundary layer heights (BLH). If the BLH was lower, that should affect all pollutants equally, right? In addition, a significant change in POA fraction, between seasons, was only observed at the urban site, not in the suburban.

Reply: The idea behind the statement cited by the reviewer is, that while POA is affected by boundary layer height since it is produced locally and mixed only within the boundary layer, transported aerosols (like sulphate or LV-OOA) are not limited to the boundary layer height because during the longer transport time also other mixing processes contribute to their dispersion within the atmosphere. Therefore, changes in boundary layer height will not influence the concentrations of transported aerosols, contrary to those of POA.

To make this clearer we added a short statement to the revised manuscript (P11, L20-24): "Also the relative fraction of sulphate could be lower in Oct/Nov due to an enhanced contribution of primary particles (Table 3) as a consequence of lower boundary layer heights (which limits the dilution of locally produced aerosols while transported aerosols are also diluted by other processes) and, potentially, higher emission strength of local sources during the colder season."

Regarding the "POA fraction": The reviewer is right that for the suburban site, the relative fraction of POA-related aerosol in the 2014 measurement (spring) is only slightly lower compared to fall, however, the absolute concentrations of POA-related aerosols are clearly lower. Therefore we changed the term "enhanced fraction" to "enhanced contribution" (see cited text in the previous paragraph).

10) P11, L22-25: The authors say that absolute concentrations of suburban home-made PM1 are larger than urban, probably due to meteorological conditions. Right. However, note that in percentage the urban is larger than the suburban (70 and 47%). In reality, home-made PM1 is smaller in absolute concentrations at the suburban because all PM1 is ~2 times smaller. Thus, the meteorological conditions are the main reason for such decrease in PM1.

The home-made PM1 decrease is just following the PM1 trend. It seems to me that the use of absolute concentrations only, can slightly misguide the reader.

Reply: We agree with the reviewer that due to the large differences in total $PM_1$ opposite trends between the two measurement sites can be observed for the absolute and relative contributions of home-made aerosol. Since the transported aerosol strongly depends on air mass origin we focused for the discussion of the home-made aerosol on the absolute values only. However, we agree that this could misguide the reader. Therefore, we added the information about the relative concentrations in the revised manuscript (P12, L18-19): "However, due to the large differences in total $PM_1$, the relative contribution of home-made $PM_1$ was higher at the urban location."

11) P13, L6-8: It would be very kind of the authors to point out in here which minerals were identified in Silvia Nara, personal communication 2016, since this communication is not easily available.

Reply: We completely agree with the referee that this information is not easily available and therefore added it to the revised manuscript (P13, L32-P14, L3). Furthermore, we are now able to replace the personal communication with a reference to a manuscript in preparation:

"PIXE analysis of 1-hour filter samples confirmed a significant increase of mineral dust concentrations (i.e. Na, Mg, Si, Al, Ti, K, Ca, Fe; Nava et al., 2012) at ground level during the identified dust periods (Barnaba et al., in preparation 2016)."

12) P18, L16-20: In fact, what Jimenez et al. (2009) state is that the larger dynamic range in solar radiation and air temperature may cause the different OOAs.

Reply: Thanks for the comment. We have clarified this in the revised manuscript (P19, L26-30):

"This is typically only observed during summer conditions, when the dynamic range of temperature, ozone concentration and solar radiation is large, which is assumed to be the main driving force for the variability of the OOA volatilities (Jimenez et al., 2009)."

13) P19, L23-28: Those last sentences are slightly confusing and seem to contradict the "home-made vs advected" section of this study. That is, in the "home-made vs advected" section nitrate was considered home-made. Here it is said that nitrate is north-east advected. Please clarify. Why are those sections separated if they complement one another?

Reply: Thank you for pointing this out. Actually, there is no contradiction but simply unfortunate wording in this sentence. We did not mean that nitrate is long-range transported from north-easterly directions, but that $NO_3$ was elevated when wind was from this direction, indicating sources for this aerosol component being located preferentially in this direction. We re-worded the respective sentence in order to make this clearer (P21, L6-8):

"Elevated NH$_4$ and SO$_4$ concentrations were mainly measured during times with south-westerly wind direction, whereas for NO$_3$ rather an increase for north-easterly directions was observed."

14) P21, L7-11: I'm surprised BBOA is still so significant in late spring. Unfortunately, the authors did not provide the MS for the reader. Are the two BBOAs (summer-like and winter-like) similar? If one BBOA was more related to agricultural and the other to agricultural and domestic heating, I'd expect to see some differences among them, was any observed? Also with respect to external tracers.

Reply: We observed agricultural burning activities during both, the late autumn and the late spring measurements. Due to the relatively mild conditions also during the Oct/Nov 2013 measurements, we do not expect strong contributions from domestic heating also during this season. To make this point clearer we added the following statement to the revised manuscript: (P21, L27-29)

"However, due to the moderate temperatures also during the Oct/Nov 2013 measurements, we do not expect strong contributions from domestic heating."

Furthermore, we have included the BBOA spectra and time series (also of the external tracers) in the new supplement. The spectra of BBOA measured during the two different seasons show clear differences, but it was not possible to identify any relations to different BBOA sources or transformation processes. Actually, for DIAPASON2013 the final PMF solution contained two factors indicating BBOA. We examined both BBOA factors in order to identify indications of different sources/processes. Since the time series of both BBOA factor spectra were quite similar (see Fig. S7) and the most dominant event of biomass burning (see P21, L31-33) was included in both BBOA factor time series, we are confident that both BBOA factors represent aerosol from the same sources and are only the result of splitting within the PMF analysis. This is also presented in the supplementary information.

15) P24, L1-4: Very interesting! The MS of CSOA identified by this study look quite different from the previous (Fröhlich et al., 2015 and Faber et al., 2013), with prominent fragments at m/zs 84. Also the N-containing fragment at m/z 42 look prominent and present in NIST MS database. Why this fragment was not mentioned in this study? Was it because of interference from other fragments in the same nominal m/z?

Reply: We are grateful to the referee for this very interesting and helpful comment. In fact, the N-containing ion at m/z 42 ($C_2H_4N^+$) shows very similar time series ($R^2>0.8$) to the CSOA factor as well as to the here proposed nicotine marker ion $C_5H_{10}N^+$. This ion $C_2H_4N^+$ is significantly more prominent in the CSOA spectra (f$C_2H_4N^+$ =1-2%) compared to all the other PMF factor spectra (f$C_2H_4N^+$ = 0.1-0.6%). However, even in the DIAPASON data, which are not affected by cigarette emissions, the variation of the concentration of this ion is quite strong, showing that this ion is not as specific to cigarette emissions as the ion at m/z 84. Therefore, it is not easy to define a criterion (e.g. a detection limit based on the POA concentration) for this ion in order to identify cigarette smoke affected data.

However, the combination of information from both ions strengthens the association of a PMF factor to cigarette emissions. Therefore, we have provided some information on that ion and a discussion of its applicability as a CSOA tracer in the revised manuscript (P25, L11; P25, 24-29; P27, L18-25; P28, L1-6).

16) P24, L5-6: Just a comment: when comparing SV-OOA and LV-OOA MS from the AMS database (Ng et al., 2010), also a R2=0.65 is obtained, and those are very different things! Thus, I'd be very careful in comparing CSOA obtained in this study to previous ones. I mean, there is no need to try to make them look similar, since the current study seems to have identified a much clearer CSOA factor.

Reply: Thank you for this hint. We completely agree that an $R^2$ of 0.65 can result from the comparison of very different types of mass spectra. The point of presenting this information is rather to show the variability that can occur in the identification of CSOA factors in different data sets.

17) P25, L3: The authors used C5H10N+ fragment as a marker for CSOA. Since the measurements were performed with and HR-ToF-AMS, which provides size distribution data. Did the authors try to identify the size distribution of the CSOA using this same marker? Or perhaps just UMR m/z 84, 42?

Reply: Thanks to the referee for this suggestion. We did not save HR size distribution data, so it is not possible to extract the ion's size distributions separately. Unfortunately also the signal-to-noise ratio of the individual UMR size distributions for m/z 84 and 42 is not sufficient to provide distinct information on the CSOA particle sizes. Therefore we did not further analyse them for this purpose.

18) P18, L4: The right citation is Alfarra et al., 2004 (Characterization of urban and rural organic particulate in the Lower Fraser Valley using two Aerodyne Aerosol Mass Spectrometers).

Reply: Thanks for the comment. We have changed this reference in the revised manuscript.

19) P18, L5-11: This sentence is too long and difficult to read.

Reply: Thank you for this hint. We replaced the original sentence: "Under conditions where sufficient freshly oxidised organic aerosol is available in the ambient air, PMF can separate the OOA into two factors associated with mass spectra containing different relative fractions of *m/z* 44 and *m/z* 43 (f44 and f43, ratio of *m/z* 44 and *m/z* 43 signal, respectively, to the total signal of organics): (I) A less oxidised, fresher, more locally produced semi-volatile OA (SV-OOA) associated with higher f43, and (II) a stronger oxidised, more aged low-volatile OA (LV-OOA) associated with higher f44 (Ng et al., 2010)."

with: "Under conditions where sufficient freshly oxidised organic aerosol is available in the ambient air, PMF can separate the OOA into two factors. These factors differ in their relative fractions of *m/z* 44 and *m/z* 43 (f44 and f43, ratio of *m/z* 44 and *m/z* 43 signal, respectively, to the total signal of organics), which reflects their different degrees of oxidation. The factor associated with higher f43 indicates a less oxidised, fresher, more locally produced semi-volatile OA (SV-OOA), whereas a higher f44 indicates a stronger oxidised, more aged low-volatile OA (LV-OOA) (Ng et al., 2010)."

20) P20, L14: Probably the ":" should be replaced by a ".".

Reply: When replacing the ":" by a "." the meaning would be changed. Since the original version was apparently not clear, we changed it to:

"This event was used during the identification of the PMF solution since only a factor including this event could be considered to be attributed to biomass burning emissions."

21) P23, L13: Replace ":" by ".".

Reply: The referee's suggestion was applied in the revised manuscript.

**Response to anonymous referee #3**

We thank anonymous referee #3 for his/her helpful comments.

**General comments:**

This paper describes measurements from campaigns in and around Rome during different seasons. It predominantly considers HR-AMS data but also incorporates estimates of Saharan dust convection and aerosol size distribution. It offers valuable insights relating to the impact of different sources on PM1 in urban areas in Europe and builds on my previous studies. It makes particular advances in the consideration of cigarette smoke and its measurement using HR-AMS. As with many short seasonal campaigns it is difficult to draw firm conclusions due to the large variation in meteorological conditions, however, the authors have contrasted different seasons without over-interpreting.

**Specific comments:**

Comment: Specific Comments Labelling NO3 as 'home made' (pg 11 line 16). Although the formation of NO3 from gaseous precursors occurs over a short time scale and will be partially 'home made', it is affected over large distances and a portion is likely to be advected. This will impact on your % estimates and should be discussed and altered appropriately.

Reply: Thank you for pointing this out. We completely agree that nitrate can also be transported and consequently contributing to the "advected" aerosol. Since the total nitrate fraction is very low (4-7%) this will result only in a small error in the total percentage of

"advected" and "home-made" aerosol. We now discuss this briefly in the revised manuscript (P12, L7-13):

"While nitrate is formed quickly and thus strongly contributes to the "home-made" aerosol fraction, it can also be transported. The resulting fraction of "home-made" might therefore be slightly biased high, however by a few percent at most due to the small total contribution of nitrate to $PM_1$. For the POPE2013 measurements OOA shows only poor correlations with $SO_4$, but slightly better with $NO_3$, suggesting either a local contribution of OOA or the transport of $NO_3$. Whatever the case may be, it will contribute to the error of the estimation."

Comment: Two modes in particle size distribution (pg 14 line 1). The second 'peak' in size distribution at 0.6 um may be an influence of reduced counting efficiency to aerosols in this size range.

Reply: We agree that for the first channel of the APS reduced counting efficiency could result in biased results. However, this should not be the case for the larger size channels. Since the second peak around 0.6 µm distributes over several size channels and since we also observe increased concentrations in this size range in the OPC data and increased $PM_1$ concentrations in the EDM during the dust event, we are quite confident that this mode is not only an artefact due to reduced counting efficiency.

Comment: Bivariate polar plots (Fig 8) NH4 is reported for 2013 but not 2014.

Reply: The polar plot of $NH_4$ for DIAPASON2014 was added to the revised manuscript. It was only omitted in the first version of the manuscript in order to avoid graphical unbalance.

We also added a short discussion on it (P20, L29-32):

"The polar plot of $NH_4$ shows a hot spot at low wind speeds in northern direction, which is also reflected in the patterns of SV-OOA and $NO_3$. Increased $NH_4$ concentrations are also observed at higher wind speeds in south-westerly direction, agreeing with the polar plot patterns of LV-OOA and $SO_4$."

Comment: Technical Comments pg 11 line 4 'lowered' should read 'lower'

Reply: Done, thank you.

Comment: pg 12 line 3 'always was' should read 'was always'

Reply: Done, thank you.

Comment: Table 5 'PM10 in PM1' should read 'PM1 in PM10'

Reply: Table 5 was changed. We now present the absolute $PM_1$ concentrations and the contribution of $PM_1$ to $PM_{10}$.

Comment: Pg 17 line 19-20 'probably less precursors are' should read 'lower concentrations of precursors are probably' and 'more precursors' should read higher concentrations of precursors

Reply: Thank you for pointing this out. We changed: "At this measurement location, probably less precursors are available than in central Rome, but more precursors than at a remote regional background site such as in the study of Costabile…"

to: "At this measurement location, probably lower concentrations of precursors are available than in central Rome, but higher concentrations than at a remote regional background site such as in the study of Costabile…".

Comment: Pg 19 line 26 'aging level' should read 'degree of aging'

Reply: We replaced: "Based on the polar plot characteristics no consistent trend indicating the aging level, the source or the formation process…"

[revised manuscript text omitted]